# Bayesian Domain Invariant Learning via Posterior Generalization of Parameter Distributions

## Abstract

Domain invariant learning aims to learn models that extract invariant features over various training domains, resulting in better generalization to unseen target domains. Recently, Bayesian Neural Networks have achieved promising results in domain invariant learning, but most works concentrate on aligning features distributions rather than parameter distributions. Inspired by the principle of Bayesian Neural Network, we attempt to directly learn the posterior distribution of network parameters given domain invariant information. We first propose a theorem to show that the invariant posterior of parameters can be implicitly inferred by aggregating posteriors on different training domains. Our assumption is more relaxed and allows us to extract more domain invariant information. We also propose a simple yet effective method, named PosTerior Generalization (PTG), that can be used to estimate the invariant parameter distribution. PTG fully exploits variational inference to approximate parameter distributions, including the invariant posterior and the posteriors on training domains. Furthermore, we develop a lite version of PTG for widespread applications. PTG shows competitive performance on various domain generalization benchmarks on DomainBed. Additionally, PTG can use any existing domain generalization methods as its prior, and combined with previous state-of-the-art method the performance can be further improved. Code will be made public.

## 1 Introduction

Distribution shift is a fundamental yet challenging problem for machine learning (Quinonero-Candela et al., 2008; Muandet et al., 2013). The common assumption of independent and identically distributed data is essential for applying the networks learned from training data to test data. However, this assumption may not hold in real-world scenarios. For example, a self-driving system may be invalid in remote districts(Li et al., 2018d; Liang et al., 2018). Therefore, it's a hot topic that how to generalize a model to out-of-distribution test datasets.

Domain generalization (DG) is a solution to distribution shift (Zhou et al., 2021a; Gulrajani & Lopez-Paz, 2020). DG usually take several training domains to train a model that generalize well on unseen test domains (Zhou et al., 2021b; Li et al., 2018b). One of the mainstream research interests in DG is Domain invariant learning (DIL)(Muandet et al., 2013; Ilse et al., 2020; Nguyen et al., 2021). Since deep neural networks (DNN) are usually trained in an end-to-end, black-box liked way, they may fail to distinguish between informative features and unrelated features. For example, in Colored MNIST recognition task (Arjovsky et al., 2019), DNNs may classify digits by color rather than shape. DIL aims to extract invariant features that shared by different domains, so the disturbance from domain specific background features will be reduced. Since domain invariant features may contain more valuable information, DIL is widely acknowledged as an effective DG method.

Uncertainty is also an important consideration for out-of-distribution generalization (Li et al., 2022b; Qiao & Peng, 2021; Upadhyay et al., 2021). Traditional DNNs are usually optimized by maximum likelihood estimation, which ignores model uncertainty and data uncertainty. Researches have validated that common DNNs are overconfident in their predictions, especially for out-of-distribution data (Guo et al., 2017; Hein et al., 2019; Daxberger & Hernández-Lobato, 2019). Bayesian neural

network (BNN) is a well-studied approach that good at uncertainty estimation (Blundell et al., 2015; Jospin et al., 2022; Kristiadi et al., 2020). BNN aims to learn the posterior distributions of parameters to represent uncertainty. Some recent works have applied BNN in DG. Xiao et al. (2021) estimate domain invariant features and classifiers by BNN, and minimize the distributional discrepancy across different domains. Liu et al. (2021) propose a novel variational Bayesian inference framework to enforce the conditional distribution alignment via the prior distribution matching in a latent space, which also takes the marginal label shift into consideration with posterior alignment.

However, in most Bayesian domain generalization methods, BNNs are treated as a tool rather than being fully explored from the perspective of their principle: the posterior distribution of parameters. DIL learn domain invariant features by adversarial learning (Li et al., 2018c; Shao et al., 2019; Li et al., 2018b), direct alignment (Li et al., 2020; Xiao et al., 2021) or other methods. From the perspective of Bayes, these methods indirectly change the estimate of parameters from Maximum a Posteriori (MAP) estimate given full training data distributions to MAP given domain invariant features, which we call domain invariant parameters. Inspired by this perception, we want to directly infer the posterior distribution of domain invariant parameters from complete given domains.

In this work, we propose a novel approach to obtain the posterior of parameters given domain invariant information, PosTerior Generalization (PTG). For brevity, we call the posterior of parameters given domain invariant information as domain invariant posterior. PTG aggregates the posterior of parameters on different training domains to directly infer the domain invariant posterior. Different from other DIL methods, PTG does not need to represent domain invariant information by feature distributions. To be specific, we just assume that there exists two abstract sufficient statistics: domain invariant information $\mathcal{D}^c$ and domain specific information $\mathcal{D}^v$. $\mathcal{D}^c$ and $\mathcal{D}^v$ represent all the domain invariant information and the rest information from $\mathcal{D}$, and they should be independent. With this condition, we can directly calculate the distribution of parameter posteriors given $\mathcal{D}^c$ by Bayes formula and other formulas. Given different training domains, we can treat these domains as samples and empirically approximate the specific form of posteriors given $\mathcal{D}^c$. At last, we simplify the distribution of parameters by variational inference for easy practical application.

We also give insights into PTG from the view of feature learning. Compared with simple DIL, PTG try to make predictions by domain invariant information extract from both invariant features and part of specific features. We also provide a lightweight, DNN based version PTG-Lite for further simplification. PTG can work as a post process that identifies the domain invariant parameters in its prior model and further aggregate the domain specific parameters, where the prior can be a model obtained by any DG method. We empirically evaluate PTG on DomainBed (Gulrajani & Lopez-Paz, 2020). Experiments show that PTG can bring improvements across various benchmarks. Combined with the state-of-the-art competitor(Li et al., 2017a), PTG can further improve its performance.

Our contributions can be summarized as follows:

- We introduce the analysis of parameter posterior distributions into domain generalization for the first time.
- Based on a relaxed assumption, we propose theories to infer the posteriors given domain invariant information, which allow us to extract more domain invariant information.
- We propose two simple yet effective domain generalization methods named Posterior Generalization based on our theories.
- Posterior Generalization achieves state-of-the-art performance on various benchmarks, and combined with other methods the performance can be further improved.

## 2 RELATED WORK

### 2.1 DOMAIN GENERALIZATION

Domain generalization aims to learn a generalized model by given training domains that can be applied to any unseen test domains (Blanchard et al., 2011; Zhou et al., 2021a; Gulrajani & Lopez-Paz, 2020; Wang et al., 2022). There are some DG works that require only single training domain (Wang et al., 2021; Qiao et al., 2020; Gao et al., 2022), but the use of multi training domains is still the mainstream setup (Segu et al., 2023; Wang et al., 2023; Li et al., 2022a). One basic DG approach

is empirical risk minimization (ERM), which simply minimizes the sum of empirical risks across all domains (Vapnik, 1991). Gulrajani & Lopez-Paz (2020) have shown that under a fair evaluation protocol, DomainBed, ERM can surprisingly outperform many DG methods. Other approaches include domain invariant learning (Nguyen et al., 2021; Muandet et al., 2013; Rame et al., 2022), data augmentation (Zhang et al., 2017; 2019; Kang et al., 2022), invariant risk minimization (Zhou et al., 2022; Lin et al., 2022; Arjovsky et al., 2019), meta learning (Li et al., 2018a; Shu et al., 2021) and other methods (Hu et al., 2018; Zhang et al., 2022; Rosenfeld et al., 2022).

## 2.2 Domain Invariant Learning

Domain invariant learning (DIL) is widely studied in various tasks. For example, in domain adaption (Csurka, 2017), where test data without labels are available, DIL aims to learn features that shared by both training and test domains (Zhao et al., 2019). There are theoretical guarantees that the invariant features work well on test domains (Ben-David et al., 2010). However, in DG, test domains are unavailable, so DIL only learns invariant features shared by training domains. Muandet et al. (2013) propose domain-invariant component analysis to learn an invariant transformation by minimizing the dissimilarity across domains. Zhao et al. (2020) propose an entropy regularization term to learn conditional-invariant features across all source domains. Rame et al. (2022) introduce a regularization that enforces domain invariance in the space of the gradients of the loss.

## 2.3 Bayesian Neural Network

Bayesian neural network aims to estimate the uncertainty of parameters (Blundell et al., 2015; Kristiadi et al., 2020; Jospin et al., 2022). The key idea of BNN is to estimate the posterior distributions of parameters given training data. Recently, researches have proposed several realization methods for BNN, including Variational Inference(Blundell et al., 2015), Markov chain Monte Carlo (Li et al., 2016) and Laplace Approximate (Daxberger et al., 2021; Kristiadi et al., 2021). There are also modern works that apply BNN in DG. Xiao et al. (2021) estimate the distribution of domain invariant features and classifiers and by BNN. Liu et al. (2021) propose a variational Bayesian inference framework to enforce the conditional distribution alignment and marginal label shift alignment by distribution alignment. However, most works use BNN to estimate the distributions of features or classifiers across different domains, rather than adapting BNN from the view of parameter distributions.

## 2.4 Variational Inference

Variational inference is a popular approach to train BNNs. It approximates the true posteriors by some common distributions, such as Gaussian distribution. The distance between variational distribution and the true posterior is quantified by Kullback-Leibler (KL) divergence. Blundell et al. (2015) propose a backpropagation-compatible algorithm for variational BNN training. Kristiadi et al. (2020) find it sufficient to build a ReLU network with a single Bayesian layer. Krishnan et al. propose a method to choose informed weight priors in BNN by DNN.

## 3 Proposed Method

In this section, we introduce the theory of PTG and how it works. We first give some necessary notations and claims in Section 3.1. Then, we explain the theory in Section 3.2. The algorithm implementations of PTG are shown in 3.3 and Section 3.4. At last, We explain how PTG extract domain invariant information from the view of feature learning.

## 3.1 Preliminaries

We introduce notations for our discussions. We denote an arbitrary domain by $\mathcal{D}$, and use $\{\mathcal{D}_i\}_{i=1}^N$ to represent training domains, where $N$ is the number of training domains. For easy description in the following passage, we define $\mathcal{D}$ to be the random variable that follows the joint distribution of data $X$ and labels $Y$ in a dataset (Zhou et al., 2021a), rather than a mark of domain labels or a collection of samples. We denote network parameters by $\omega$. To simplify the description, we use $p(\cdot)$ to denote the distribution of corresponding variables. For example, $p(\mathcal{D})$ means the distribution of $\mathcal{D}$.

We assume that there exist two independent sufficient statistics of each domain: domain invariant information $\mathcal{D}^c$ and domain specific information $\mathcal{D}^v$. $p(\mathcal{D}^c)$ remains constant as $\mathcal{D}$ changes, but $p(\mathcal{D}^v)$ vary. The principle behind this assumption is shown in Appendix A. We denote the domain specific information of each training domain as $\{\mathcal{D}_i^v\}_{i=1}^N$. **We do not need to assume the form of these two statistics**, while they usually exist as domain invariant and variant features (Shankar et al., 2018). Furthermore, **we do not need to specify how $\mathcal{D}^c$ and $\mathcal{D}^v$ are extracted from $\mathcal{D}$**. We can approximate the posterior distribution of parameters given $\mathcal{D}^c$, $p(\omega|\mathcal{D}^c)$, even without access to $\mathcal{D}^c$.

At last, we briefly introduce how to infer the posterior of parameters by variational inference (Blundell et al., 2015). $p(\omega|\mathcal{D}_i)$ denotes the posterior distribution of parameters given domain $\mathcal{D}_i$, and $q(\omega|\theta_i)$ denotes a variational distribution, where $\theta_i$ is the parameter of the variational distribution. We use Gaussian distribution as the variational distribution. If we train a BNN on $\mathcal{D}_i$, its loss function is:

$$\mathbb{D}_{KL}[q(\omega|\theta_i)||p(\omega|\mathcal{D}_i)] = \int q(\omega|\theta_i) log(\frac{q(\omega|\theta_i)}{p(\omega|\mathcal{D}_i)}) \, d\omega. \tag{1}$$

By simplification, the loss function is:

$$\mathbb{D}_{KL}[q(\omega|\theta_i)||p(\omega)] - \mathbb{E}_{q(\omega|\theta_i)}[log(p(\mathcal{D}_i|\omega))], \tag{2}$$

where $p(\omega)$ means the prior distribution of parameter, which is usually set to be standard Gaussian distribution. The first loss term can be seen as a regularization and the second term is the original negative log-likelihood. In practice, the second term can be empirically optimized and the first term has an explicit expression. After training, we can approximate the intractable posterior $p(\omega|\mathcal{D}_i)$ by tractable variational distribution $q(\omega|\theta_i)$.

### 3.2 BAYESIAN PRINCIPLE OF PTG

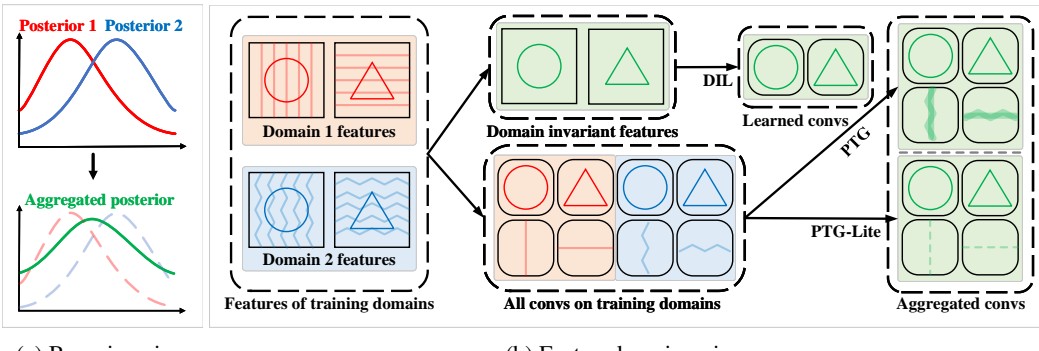

(a) Bayesian view          (b) Feature learning view

Figure 1: Illustration of PTG from Bayesian view and feature learning view. From Bayesian view, PTG aggregates posteriors on each domain to infer domain invariant posteriors. From feature learning view, PTG extracts more domain invariant information from feature. DIL aims to extract invariant features while ignoring the similar but variant features. PTG methods aim to infer the invariant parameter posteriors by different aggregation approaches (separated by gray dashed line). As a result, PTG methods can preserve the invariant information from specific features.

To train a network that can generalize on any domain, we aim to estimate the posterior of parameters given domain invariant information $p(\omega|\mathcal{D}^c)$. However, due to the unknown content of $\mathcal{D}^c$, $p(\omega|\mathcal{D}^c)$ is intractable, let alone estimation. In fact, $\mathcal{D}^c$ and $\mathcal{D}^v$ are independent, but they always exist together. We can only get $p(\omega|\mathcal{D}^c, \mathcal{D}^c)$. Nevertheless, we can infer $p(\omega|\mathcal{D}^c)$ by the following formula:

**Theorem 3.1.** *If $\mathcal{D}^c$ and $\mathcal{D}^v$ are independent, then $p(\omega|\mathcal{D}^c) = \mathbb{E}_{p(\mathcal{D}^v)}[p(\omega|\mathcal{D}^c, \mathcal{D}^v)]$*

The proof is shown in Appendix B. As a result, we can empirically estimate $p(\omega|\mathcal{D}^c)$ by sampling from $p(\mathcal{D}^v)$. Since $p(\mathcal{D}^c)$ is constant, sampling from $p(\mathcal{D}^v)$ is the same as sampling from $p(\mathcal{D})$, which is exactly $\{\mathcal{D}_i\}_{i=1}^N$. Meanwhile, $p(\omega|\mathcal{D}^c, \mathcal{D}_i^v) = p(\omega|\mathcal{D}_i)$ because $\mathcal{D}^c$ and $\mathcal{D}_i^v$ are sufficient statistics of $\mathcal{D}_i$. Considering that $p(\omega|\mathcal{D}_i)$ can be approximate by $q(\omega|\theta_i)$ via variational inference, we can approximate $p(\omega|\mathcal{D}^c)$ by:

$$p(\omega|\mathcal{D}^c) \approx \frac{\sum_{i=1}^N q(\omega|\theta_i)}{N}. \tag{3}$$

Note that **it's the mean of distributions** $q(\omega|\theta_i)$, rather than the mean of parameters $\omega$. For the convenience of realization, we keep approximating $p(\omega|\mathcal{D}^c)$ by Gaussian variational inference. The approximate expectation and variance of $p(\omega|\mathcal{D}^c)$ can be calculated by Appendix C. Therefore, we replace the true domain invariant posterior by $q(w|\theta_0)$:

$$q(w|\theta_0) = \mathcal{N}(\mu, \sigma^2) \tag{4}$$

$$\mu = \frac{\sum_{i=1}^{N} \mathbb{E}_{q(\omega|\theta_i)}[\omega]}{N} \tag{5}$$

$$\sigma^2 = \frac{\sum_{i=1}^{N} \mathbb{VAR}_{q(\omega|\theta_i)}[\omega]}{N} + \frac{\sum_{i=1}^{N} \mathbb{E}_{q(\omega|\theta_i)}[\omega^2]}{N} - \left(\frac{\sum_{i=1}^{N} \mathbb{E}_{q(\omega|\theta_i)}[\omega]}{N}\right)^2 \tag{6}$$

where $\mu$ and $\sigma^2$ are approximate expectation and variance. We give an illustration for the Bayesian view of PTG in Figure 1a.

### 3.3 IMPLEMENTATION OF PTG

Although we have made some simplifications in 3.2 to put the theory into practice, there are still many difficulties. The first problem is the **disordered dimensions of parameters**. For example, if we train two BNNs on two domains by the same method, there's no guarantee that parameters at the same position have the same function. The first convolution kernel in the first BNN mat extract foreground features and the second convolution kernel extracts background features. The opposite situation may exist in the second BNN. If we directly calculate $p(\omega|\mathcal{D}^c)$ by PTG without addressing this issue, the aggregated convolution kernels will have great variances, and their function can be hardly explained. To mitigate this problem, we should initialize the BNN on each domain by the same, well-generalized model, e.g. a BNN trained by ERM. In this way, the function of each parameter can be approximately settled, which avoids the problem of disorder to some extent.

Another problem is the **ambiguity of classifier**. Since different training domain contains different features, the distribution of classifier, i.e. the last layers in a network, may differ a lot across domains. Similarly, if we directly calculate the posterior of domain invariant classifier, some parts of the final classifier may have large variances, which can influence the interpretability or even hurt the prediction performance. Therefore, we only construct one classifier shared by different domains, and further optimize it after the aggregation of featurizers. Besides, we design the classifier to be deterministic layers for less ambiguity.

The last problem is the **dimension reorder of parameters**. Although initialization can set parameters near extreme points, if the learning rate is too large, parameters may deviate from their local minima during training, leading to the problem of disordered dimension again. As a result, the learning rate of PTG should be carefully decayed by a rate $\alpha$, such as 0.01 times the learning rate of initialization methods. To make sure the aggregated parameters can still extract meaningful features, we further update them by ERM. The algorithm of PTG is summarized as Algorithm 1

---

**Algorithm 1** PTG

---

**Input:** training domains $\{\mathcal{D}\}_{i=1}^{N}$
Initialize BNN featurizers $\{f_i(\cdot)\}_{i=0}^{N}$ and DNN classifier $f_{cls}(\cdot)$ by a DG method
**for** training iterations **do**
    **for** i=1; i⩽N; i++ **do**
        sample minibatch data $(x_i, y_i)$ from $\mathcal{D}_i$
        calculate loss by $(f_{cls}(f_i(x_i)), y_i)$ and Equation (2)
        update $f_i(\cdot)$ with $\alpha$ decayed learning rate
    **end for**
    update $f_0(\cdot)$ by Equation (4)
    merge $\{(x_i, y_i)\}_{i=1}^{N}$ to form $(X, Y)$
    calculate loss by $(f_{cls}(f_0(X)), Y)$ and Equation (2)
    update $f_0(\cdot)$ and $f_{cls}(\cdot)$ with $\alpha$ decayed learning rate
**end for**
**Output:** generalized network $f_{cls}(f_0(\cdot))$

---

### 3.4 PTG-LITE

Although PTG exploit variational inference to simplify the aggregation of posteriors, the training of BNNs and the inference of PTG are still complicated. Therefore, we further simplify PTG and propose the DNN based PTG-lite. PTG-Lite shares the same Bayesian theory with PTG, but PTG-Lite uses MAP to simplify the invariant variational distribution $q(\omega|\theta_0)$. Since we choose Gaussian distribution to be the variational distribution in PTG, the MAP estimate is exactly the expectation, so the aggregated parameters can be calculated by:

$$\theta_0 = \frac{\sum_{i=1}^{N} \mathbb{E}_{q(\omega|\theta_i)}[\omega]}{N}. \tag{7}$$

Similarly, the expectations of variational distributions on different domains $q(\omega|\theta_i)$are exactly their MAP estimates. According to Equation (2), the MAP estimate can be obtained by a maximum likelihood estimate (right) plus L2 regularization (left).

Different from PTG, PTG-Lite can't represent the uncertainty of parameters, so the domain specific parameters are not effectively aggregated or may even ruin the whole network. We study by experiments that it works better to drop out domain specific parameters than to replace them by mean values. We judge whether a parameter is domain specific by its coefficient of variation: if the coefficient of a parameter on different domains is greater than a given rate $\beta$, such as 0.1, we drop out this parameter. The algorithm of PTG-Lite is summarized as Algorithm 2

---

**Algorithm 2** PTG-Lite

**Input:** training domains $\{\mathcal{D}\}_{i=1}^{N}$
Initialize DNN featurizers $\{f_i(\cdot)\}_{i=0}^{N}$ and DNN classifier $f_{cls}(\cdot)$ by a DG method
**for** training iterations **do**
    **for** i=1; i≤N; i++ **do**
        sample minibatch data $(x_i, y_i)$ from $\mathcal{D}_i$
        calculate loss by $(f_{cls}(f_i(x_i)), y_i)$ and Equation (2)
        update $f_i(\cdot)$ with $\alpha$ decayed learning rate
    **end for**
    update $f_0(\cdot)$ by $\frac{\sum_{i=1}^{N} f_i(\cdot)}{N}$
    drop out $f_0(\cdot)$ by coefficient of variation and rate $\beta$
    merge $\{(x_i, y_i)\}_{i=1}^{N}$ to form $(X, Y)$
    calculate loss by $(f_{cls}(f_0(X)), Y)$ and Equation (2)
    update $f_0(\cdot)$ and $f_{cls}(\cdot)$ by ERM with $\alpha$ decayed learning rate
**end for**
**Output:** generalized network $f_{cls}(f_0(\cdot))$

---

### 3.5 EXPLANATION FROM FEATURE LEARNING VIEW

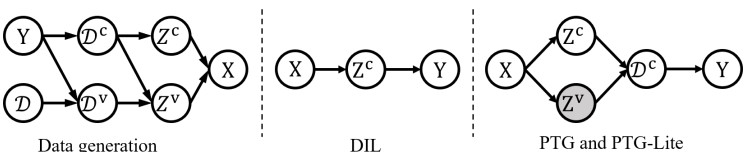

Figure 2: Casual relationships. We assume there exists domain invariant information $\mathcal{D}^c$ and domain specific information $\mathcal{D}^v$and follow the data generation assumption (left) as Rosenfeld et al. (2020). Most DIL (middle) makes inference by domain invariant features $Z^c$, which fail to provide enough invariant information. PTG methods (right) makes inference by domain invariant information directly, which is extracted from both invariant features and useful specific features. Gray node means the specific features are extracted by aggregated parameter posteriors.

Although the Bayesian principle of PTG is provided in Section 3.2, we can give a more intuitive description of how PTG works from the view of feature learning. Moreover, the relationship between our assumption, domain invariant information, and domain invariant features can be better illustrated.

As shown in Figure 2, traditional DIL makes a stronger assumption that domain invariant information exist in the form of feature maps, which may ignore some potential information that exists in specific features. In contrast, PTG directly infers the posterior distribution of parameters conditioned on the domain invariant information. And we show in the next textbf that these parameters can extract invariant information from both invariant and specific features.

There is a strong relationship between domain invariant parameters $p(\omega|\mathcal{D}^c)$ and its variation rate, details are discussed in Appendix D. During the aggregation, posteriors that differ little across domains will be replaced by similar distributions; while posteriors that differ a lot will be replaced by new distributions with large variances. Consequently, PTG keeps the invariant parameters while aggregating specific parameters into more general distributions. PTG-Lite aggregates parameters by dropping out extreme specific parameters, but some specific parameters are reserved. From this perspective, PTG is more like a post process: it further identifies the remaining domain specific parameters within a prior model, and aggregate them by general parameter distributions. We give a visualization of this process in Figure 1b, where the synthetic specific features contain significant invariant information. For easy understanding, we use a whole convolution kernel to represent domain invariant or specific parameters. In fact, the domain invariant and specific parameters are mixed up.

## 4 EXPERIMENTS

### 4.1 EXPERIMENT SETUP

**Datasets.** Following Gulrajani & Lopez-Paz (2020), we evaluate our method and comparison methods on four benchmarks: PACS (Li et al., 2017b), VLCS (Fang et al., 2013), OfficeHome (Venkateswara et al., 2017), TerraIncognita (Beery et al., 2018).

**Evaluation protocol.** We follow the training and evaluation protocol in DomainBed. We select one domain as the target domain while the rest domains are used for training. We repeat the procedure until all domains have been used as test domains. **We select models via training domain validation set** (Gulrajani & Lopez-Paz, 2020). The results that use other model selection methods are reported in Appendix F Each training domain is divided into 8:2 training/validation splits randomly, and the final result is selected according to the detection accuracy on these validation sets. We repeat $5 \times 5$ experiments for each set up, which consist of 5 different hyperparameter samples times 5 different random seeds.

**Implementation details.** We use ResNet18 (He et al., 2016) pre-trained on ImageNet (Deng et al., 2009) as the backbone networks for all models. The Results on ResNet50 are shown in Appendix G. We train a BNN by other DG methods as the initializations of PTG. PTG-Lite can directly use other DG models as its initializations. All the BN layers are frozen during training. The last FC layer is replaced by a classifier with 1024 hidden units. We also apply dropout. Models are trained using the Adam optimizer. The search space of $\alpha$ is $\{0.05, 0.1, 0.5\}$, and $\{0.05, 0.1\}$ for $\beta$. We do not use other strategies such as weight averaging (Cha et al., 2021) or ensemble learning (Li et al., 2023) to directly show the influence of PTG. More details are shown in Appendix E.

### 4.2 MAIN RESULTS

We compare PTG with the following methods: Mixup (Yan et al., 2020), CORAL (Li et al., 2017a), MMD (Sun & Saenko, 2016), IRM (Arjovsky et al., 2019), GroupDRO (Sagawa et al., 2019), CAD (Ruan et al., 2021), VREx (Krueger et al., 2021), SagNet (Nam et al., 2021), Bayes-IRM (Lin et al., 2022), Fish (Shi et al., 2022), Fishr (Rame et al., 2022), ERM, ARM (Zhang et al., 2022), SD(Pezeshki et al., 2021), and SelfReg (Koyama & Yamaguchi, 2020). We only compare with models that do not use large scale pre-training or ensemble learning.

The overall out-of-domain detection accuracies performances on four DG benchmarks are reported in Table 1. We show the full tables reporting the performance on each benchmark in Appendix I. In all experiments, PTG achieves significant performance gain against ERM-Bayesian as well as the previous best results: +1.9% in PACS, +0.2% in VLCS, +4.4% in TerraIncognita and +2.2% in average compared to the previous state-of-the-art model. BNNs are recognized to have strong generalization ability because they catch uncertainty from training data. However, we observe that although ERM-Bayesian gains improvements on PACS and OfficeHome compared to ERM, the

Table 1: **Benchmark Comparisons**. Out-of-domain classification accuracies(%) on PACS, VLCS, OfficeHome and TerraIncognita are shown. ERM-Bayesian is a BNN (Blundell et al., 2015) trained by ERM. PTG takes ERM-Bayesian as initialization. PTG-Lite takes ERM as initialization. All models are reproduced on DomainBed. We highlight the **best**, second and third results.

| Algorithm | PACS | VLCS | Office-Home | TerraIncognita | Avg |
|---|---|---|---|---|---|
| CAD | $67.4 \pm 6.2$ | $66.6 \pm 2.2$ | $26.6 \pm 9.9$ | $27.5 \pm 3.9$ | 47.0 |
| IRM | $78.9 \pm 1.2$ | $73.6 \pm 1.4$ | $49.7 \pm 4.8$ | $32.2 \pm 3.4$ | 58.6 |
| MMD | $80.8 \pm 1.5$ | $74.2 \pm 0.9$ | $58.4 \pm 0.4$ | $33.1 \pm 9.6$ | 61.6 |
| ARM | $79.2 \pm 0.9$ | $74.3 \pm 0.9$ | $56.7 \pm 0.4$ | $36.6 \pm 1.0$ | 61.7 |
| GroupDRO | $80.3 \pm 0.5$ | $73.9 \pm 0.6$ | $58.0 \pm 0.2$ | $34.8 \pm 2.2$ | 61.8 |
| VREx | $81.2 \pm 0.3$ | $74.4 \pm 1.7$ | $59.1 \pm 0.3$ | $37.4 \pm 0.5$ | 63.0 |
| Bayes-IRM | $81.1 \pm 0.4$ | $74.7 \pm 1.3$ | $59.3 \pm 0.3$ | $38.9 \pm 1.1$ | 63.5 |
| Mixup | $79.4 \pm 0.1$ | $74.4 \pm 0.8$ | $60.0 \pm 0.5$ | $40.3 \pm 1.4$ | 63.5 |
| Fishr | $81.2 \pm 0.9$ | $75.4 \pm 0.4$ | $59.1 \pm 1.1$ | $40.1 \pm 0.7$ | 64.0 |
| SD | $80.2 \pm 1.0$ | $75.0 \pm 0.9$ | $\mathbf{62.2 \pm 0.3}$ | $38.6 \pm 3.3$ | 64.0 |
| SagNet | $81.2 \pm 0.9$ | $75.8 \pm 0.4$ | $60.2 \pm 1.1$ | $39.3 \pm 2.1$ | 64.1 |
| SelfReg | $81.8 \pm 1.1$ | $75.3 \pm 1.0$ | $61.2 \pm 0.4$ | $38.2 \pm 2.4$ | 64.1 |
| Fish | $80.7 \pm 0.3$ | $75.9 \pm 0.5$ | $61.2 \pm 0.4$ | $39.0 \pm 1.2$ | 64.2 |
| CORAL | $81.2 \pm 0.5$ | $75.4 \pm 0.6$ | $61.9 \pm 0.2$ | $38.7 \pm 3.1$ | 64.3 |
| ERM | $79.8 \pm 1.2$ | $75.7 \pm 0.2$ | $58.9 \pm 1.0$ | $41.7 \pm 1.5$ | 64.0 |
| PTG-Lite | $83.0 \pm 0.3$ | $75.9 \pm 0.3$ | $60.9 \pm 0.0$ | $\mathbf{44.9 \pm 0.4}$ | 66.2 |
| ERM-Bayesian | $81.3 \pm 0.3$ | $74.0 \pm 0.7$ | $59.2 \pm 0.7$ | $40.9 \pm 0.6$ | 63.9 |
| PTG | $\mathbf{83.7 \pm 0.1}$ | $\mathbf{76.1 \pm 0.5}$ | $61.6 \pm 0.4$ | $44.7 \pm 1.2$ | $\mathbf{66.5}$ |

average accuracy drops, which means directly applying BNN into DG task brings little benefit. However, the outstanding performance of PTG shows that Bayesian learning is still a promising approach to solve DG problem, as long as we explore its full potential.

Besides PTG, we find that PTG-Lite also achieves good performance. PTG-Lite achieves gains against ERM by: +3.2% in PACS, +0.2% in VLCS, +2.0% in OfficeHome, +3.2% in TerraIncognita and +2.2% in average. This may indicate that the parameters of ERM is already enough to extract necessary domain invariant features, but it also extracts some unnecessary features that may harm the generalization on target domains. Please refer to Section 5 for more details.

### 4.3 COMBINATION WITH OTHER METHODS

PTG needs an initialization network that trained by other DG methods. For a fair comparison, we use ERM as the initialization methods in Table 1 since ERM introduces no additional DG training strategy. However, PTG can take any other DG model as its initialization, as long as the backbone structure is not changed. Here, we combine PTG with ERM and the previous state-of-the-art model CORAL to further show the power of PTG. Similarly, we initialize and further train BNNs by CORAL, and use these BNNs to initialize PTG. More combinations are shown in Appendix H

Results are presented in Table 2. CORAL shows better performances than ERM with +0.3% average out-of-domain accuracy gain. By combining PTG and CORAL, the performances are consistently improved by 2.3% over CORAL in average. We observe that PTG can improve the accuracies across almost all experimental setups, including different prior methods, different benchmarks and different domains. We attribute this phenomenon to the dependency of the theorems of PTG and former DG methods. PTG focuses on the distribution of parameters alone, while there is no restriction about feature maps. Therefore, we believe that PTG can be easily combined with other DG methods and may get comprehensive improvements.

## 5 DISCUSSIONS AND LIMITATIONS

**Difference between PTG and PTG-Lite.** Instead of Bayesian and non-Bayesian, the major difference between PTG and PTG-Lite roots in the aggregation process. As shown in Section 3.5, the aggregation procedure of PTG can be regarded as making addition: we keep the domain invariant parameters

Table 2: **Combination with other methods.** We combine PTG with previous state-of-the-art method and report the performance on each benchmark. Each experiment is repeated 5 times.

| Dataset | Algorithm | Test Domains | | | | Avg |
|---|---|---|---|---|---|---|
| | | **A** | **C** | **P** | **S** | |
| PACS | ERM | $79.0 \pm 0.2$ | $74.3 \pm 1.7$ | $94.4 \pm 0.7$ | $71.4 \pm 2.3$ | 79.8 |
| | PTG | $82.6 \pm 0.1$ | $77.0 \pm 0.3$ | $94.7 \pm 0.4$ | $80.6 \pm 0.5$ | 83.7 |
| | CORAL | $79.6 \pm 1.0$ | $75.7 \pm 0.3$ | $94.5 \pm 0.1$ | $75.2 \pm 0.5$ | 81.2 |
| | CORAL-PTG | $82.8 \pm 0.7$ | $77.9 \pm 0.6$ | $94.9 \pm 0.2$ | $82.5 \pm 0.3$ | 84.5 |
| | | **C** | **L** | **S** | **V** | |
| VLCS | ERM | $96.0 \pm 0.3$ | $63.4 \pm 1.1$ | $70.6 \pm 1.2$ | $72.8 \pm 1.2$ | 75.7 |
| | PTG | $97.3 \pm 0.2$ | $64.6 \pm 1.2$ | $68.6 \pm 0.5$ | $73.9 \pm 0.5$ | 76.1 |
| | CORAL | $95.3 \pm 1.2$ | $64.6 \pm 0.9$ | $70.3 \pm 0.7$ | $71.4 \pm 0.2$ | 75.4 |
| | CORAL-PTG | $97.1 \pm 0.6$ | $64.8 \pm 1.4$ | $70.4 \pm 0.2$ | $71.9 \pm 0.8$ | 76.0 |
| | | **A** | **C** | **P** | **R** | |
| OfficeHome | ERM | $51.0 \pm 1.6$ | $46.8 \pm 1.4$ | $68.3 \pm 1.2$ | $69.5 \pm 1.5$ | 58.9 |
| | PTG | $55.3 \pm 0.5$ | $50.8 \pm 0.2$ | $69.7 \pm 0.3$ | $70.6 \pm 0.4$ | 61.6 |
| | CORAL | $55.4 \pm 0.9$ | $48.7 \pm 0.2$ | $71.2 \pm 0.6$ | $72.2 \pm 0.3$ | 61.9 |
| | CORAL-PTG | $57.2 \pm 1.2$ | $50.3 \pm 0.8$ | $71.6 \pm 0.5$ | $73.9 \pm 0.8$ | 63.3 |
| | | **L100** | **L38** | **L43** | **L46** | |
| TerraIncognita | ERM | $49.5 \pm 3.1$ | $32.1 \pm 3.0$ | $50.8 \pm 0.1$ | $34.2 \pm 0.4$ | 41.7 |
| | PTG | $48.6 \pm 0.8$ | $40.7 \pm 0.3$ | $52.7 \pm 0.3$ | $36.8 \pm 0.4$ | 44.7 |
| | CORAL | $45.4 \pm 5.2$ | $27.3 \pm 6.3$ | $51.4 \pm 2.1$ | $30.7 \pm 0.9$ | 38.7 |
| | CORAL-PTG | $46.0 \pm 2.2$ | $36.1 \pm 1.7$ | $52.2 \pm 0.7$ | $33.5 \pm 0.6$ | 42.0 |

while replace the domain specific parameters by general distributions. However, PTG-Lite is making subtraction: we drop the domain specific parameters directly. Both PTG and PTG-Lite can improve performance, which implies two possible research directions: (1) DG methods can benefit from some useful domain specific parameters; (2) Many DG methods already learn enough domain invariant parameters, but there are still some harmful domain specific parameters.

**PTG depends on initialization and the number of training domains.** From feature learning view, PTG is a post-procedure that refines the parameters of its prior network. Consequently, if the prior model fails to learn enough domain invariant parameters, PTG also fails. Besides, PTG estimates the invariant posterior empirically, so the number of training domain can influence the estimation reliability. We recommend the number of training domain to be 3 at least. However, we find in Appendix F that even if trained by only 2 training domains, PTG is still competitive.

**PTG is not memory efficient.** Although we have made many simplifications, the parameters on different domains have to be loaded to compute the mean and variance of parameter distributions. Besides, a BNN doubles the parameter amount of a DNN. We recommend the memory to be over 24G. Meanwhile, the training procedure of BNN is also memory consuming. However, even if we sacrifice the performance to save memory, as shown in Appendix G PTG is still competitive. Furthermore, PTG just needs a few iterations (50 iterations, 1.4 epochs), so the computational costs are low.

## 6 CONCLUSION

In this paper, we introduce the analysis of parameter posterior distributions into Domain Invariant Learning for the first time. We theoretically show how to infer the domain invariant posterior without access to the domain invariant information condition Our relaxed assumption allow us to extract more domain invariant information. We propose a new DIL method named PTG, and explained its principles form both Bayesian view and feature learning view. Furthermore, we develop a lite, non-Bayesian version of PTG for widespread applications. The extensive experiments can show the promising performance of PTG. Besides, the combination of PTG and other methods may bring comprehensive improvements. We hope that our research promotes new research directions of examining the distributions of parameters for domain generalization.

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

# Appednix

## A PRINCIPLE OF THE ASSUMPTION THAT $\mathcal{D}^c$ AND $\mathcal{D}^v$ EXIST AND ARE INDEPENDENT.

We do not assume the form of existence of $\mathcal{D}^c$ and $\mathcal{D}^v$, or how they can be obtained. They are two abstract statistics that contain all the domain invariant information and the rest information of $\mathcal{D}$. We only assume their existence and independence. Based on our assumption, the domain invariant features are part of $\mathcal{D}^c$ or can be further inferred from $\mathcal{D}^c$.

We can provide the rationality of our assumption. (1)If $\mathcal{D}^c$ didn't exist, which means there were no invariant information among domains, then Domain Generalization would have no solution. (2)If $\mathcal{D}^v$ didn't exist, which means there were only invariant information among domains, then there would be no need for further generalization. (3)If $\mathcal{D}^c$ and $\mathcal{D}^v$ exist but could not be separated, then the domain invariant features should always contain specific information (otherwise, part of $\mathcal{D}^c$ should be independent of $\mathcal{D}^v$, and we can take it as the true $\mathcal{D}^c$), and domain invariant learning would have no solution.

For example, if consider a DG task on dog images, $\mathcal{D}$ means a domain of dogs, and $\mathcal{D}_i$ may refer to a specific kind of dog, like golden retriever. $\mathcal{D}^c$ means the common information that all dogs share, such as the overall body shape, the facial patterns, the teeth and so on. $\mathcal{D}_i^v$ means the specific information that golden retriever have, such as the color, the hair length. As stated before, we don't care how the information is represented or how to get it, we just assume that it exists. We can directly estimate the posterior parameter distributions that use the common information for recognition. This is why we claim that our assumption is more relaxed than domain invariant features.

However, $\mathcal{D}^c$ is determined by the task, rather than our definition. In the dog image example, it's the nature of dog and the nature of photography that determine what information is invariant. There should be invariant information, we can tell part of the invariant information, but it's intractable to provide a rigorous representation of the whole invariant information.

## B PROOF OF THEOREM 3.1

If $\mathcal{D}^c and \mathcal{D}^v$ are independent, then $p(\omega|\mathcal{D}^c) = \mathbb{E}_{p(\mathcal{D}^v)}[p(\omega|\mathcal{D}^c, \mathcal{D}^v)]$

*Proof.*

$$\mathbb{E}_{p(\mathcal{D}^v)}[p(\omega|\mathcal{D}^c, \mathcal{D}^v)] \tag{8}$$

$$= \int p(\mathcal{D}^v)p(\omega|\mathcal{D}^c, \mathcal{D}^v)\, d\mathcal{D}^v \tag{9}$$

$$= \int p(\mathcal{D}^v)\frac{p(\omega, \mathcal{D}^c, \mathcal{D}^v)}{p(\mathcal{D}^c, \mathcal{D}^v)}\, d\mathcal{D}^v. \tag{10}$$

Since $\mathcal{D}^c$ and $\mathcal{D}^v$ are independent,

$$\int p(\mathcal{D}^v)\frac{p(\omega, \mathcal{D}^c, \mathcal{D}^v)}{p(\mathcal{D}^c, \mathcal{D}^v)}\, d\mathcal{D}^v \tag{11}$$

$$= \int p(\mathcal{D}^v)\frac{p(\omega, \mathcal{D}^c, \mathcal{D}^v)}{p(\mathcal{D}^c)p(\mathcal{D}^v)}\, d\mathcal{D}^v \tag{12}$$

$$= \int \frac{p(\omega, \mathcal{D}^c, \mathcal{D}^v)}{p(\mathcal{D}^c)}\, d\mathcal{D}^v \tag{13}$$

$$= \frac{p(\omega, \mathcal{D}^c)}{p(\mathcal{D}^c)} \tag{14}$$

$$= p(\omega|\mathcal{D}^c) \tag{15}$$

$$\square$$

## C    MATHEMATICAL DERIVATION FOR THE EXPECTATION AND VARIANCE OF $p(\omega|\mathcal{D})$

We use $f_{p(\cdot)}$ to represent the density function of corresponding distribution.

$$f_{p(\omega|\mathcal{D}^c)} \approx \frac{\sum_{i=1}^{N} f_{q(\omega|\theta_i)}}{N} \tag{16}$$

$$\mathbb{E}_{p(\omega|\mathcal{D}^c)}[\omega] = \int x f_{p(\omega|\mathcal{D}^c)}(x)\, dx \tag{17}$$

$$\approx \int x \frac{\sum_{i=1}^{N} f_{q(\omega|\theta_i)}(x)}{N}\, dx \tag{18}$$

$$= \frac{\sum_{i=1}^{N} \mathbb{E}_{q(\omega|\theta_i)}[\omega]}{N} \tag{19}$$

$$\mathbb{E}_{p(\omega|\mathcal{D}^c)}[\omega^2] = \int x^2 f_{p(\omega|\mathcal{D}^c)}(x)\, dx \tag{20}$$

$$\approx \int x^2 \frac{\sum_{i=1}^{N} f_{q(\omega|\theta_i)}(x)}{N}\, dx \tag{21}$$

$$= \frac{\sum_{i=1}^{N} \mathbb{E}_{q(\omega|\theta_i)}[\omega^2]}{N} \tag{22}$$

$$\mathbb{VAR}_{p(\omega|\mathcal{D}^c)}[\omega] = \mathbb{E}_{p(\omega|\mathcal{D}^c)}[\omega^2] - \mathbb{E}_{p(\omega|\mathcal{D}^c)}[\omega]^2 \tag{23}$$

$$\approx \frac{\sum_{i=1}^{N} \mathbb{E}_{q(\omega|\theta_i)}[\omega^2]}{N} - \left(\frac{\sum_{i=1}^{N} \mathbb{E}_{q(\omega|\theta_i)}[\omega]}{N}\right)^2 \tag{24}$$

$$= \frac{\sum_{i=1}^{N} (\mathbb{E}_{q(\omega|\theta_i)}[\omega^2] - \mathbb{E}_{q(\omega|\theta_i)}[\omega^2])}{N} + \frac{\sum_{i=1}^{N} \mathbb{E}_{q(\omega|\theta_i)}[\omega^2]}{N} - \left(\frac{\sum_{i=1}^{N} \mathbb{E}_{q(\omega|\theta_i)}[\omega]}{N}\right)^2 \tag{25}$$

$$= \frac{\sum_{i=1}^{N} \mathbb{VAR}_{p(\omega|\theta_i)}[\omega]}{N} + \frac{\sum_{i=1}^{N} \mathbb{E}_{q(\omega|\theta_i)}[\omega^2]}{N} - \left(\frac{\sum_{i=1}^{N} \mathbb{E}_{q(\omega|\theta_i)}[\omega]}{N}\right)^2 \tag{26}$$

## D    RELATIONSHIP BETWEEN $p(\omega|\mathcal{D}^c)$ AND ITS VARIATION RATE.

Since $\mathcal{D}_i^c$ follows the same distribution among domains, $p(\omega|\mathcal{D}_i^c)$ also follows the same distribution among domains. On the other hand, $\mathcal{D}_i^v$ follows different distributions among domains, so $p(\omega|\mathcal{D}_i^v)$ also follows different distributions. From another perspective, in Bayesian Neural Networks, larger variance in a parameter implies higher uncertainty. In DG context, uncertainty mainly comes from the difference between domains, so parameters that change a lot among domains(high variance) are more likely to extract domain specific features.

## E    EXPERIMENT SETUP

**Datasets.** Following Gulrajani and Lopez-Paz (Gulrajani & Lopez-Paz, 2020), we evaluate our method and comparison methods on four benchmarks: PACS (Li et al., 2017b) containing 9,991 images of 7 classes across 4 domains {photo, art, cartoon, sketch}, VLCS (Fang et al., 2013) containing 10,729 images of 5 classes across 4 domains {VOC2007, LabelMe, Caltech101, SUN09}, OfficeHome (Venkateswara et al., 2017) containing 15,588 images of 65 classes across 4 domains {art, clipart, product, real}, TerraIncognita (Beery et al., 2018) containing 24,788 images of 10 classes across 4 domains {L100, L38, L43, L46}.

**Evaluation protocol.** For a fair comparison, we follow the training and evaluation protocol in DomainBed. We select one domain as the target domain while the rest domains are used for training. We repeat the procedure until all domains have been used as test domains. We select models via training domain validation set (Gulrajani & Lopez-Paz, 2020). Each training domain is divided into 8:2 training/validation splits randomly, and the final result is selected according to the detection

accuracy on these validation sets. We repeat $5 \times 5$ experiments for each set up, which consist of 5 different hyperparameter samples times 5 different random seeds. We select the best hyperparameter and report the mean and standard deviation of test domain classification accuracies from 5 random runs.

**Implementation details.** We use ResNet18 (He et al., 2016) pre-trained on ImageNet (Deng et al., 2009) as the backbone networks for all models. However, PTG and PTG-Lite need pre-trained DG models as their initializations. The initializations of PTG should be BNNs. To reduce the computation cost, we specify the prior of BNNs by DG trained DNNs (Krishnan et al., 2020) and further train the BNN in the same way. Then, the BNNs are used as the initializations of PTG. PTG-Lite can directly use the DNNs trained by other DG methods as its initializations. All the BN layers are frozen during training. The last FC layer is replaced by a classifier with 1024 hidden units. We also apply dropout where the dropout rate is selected by DomainBed. Models are trained using the Adam optimizer. The search space of decay rate $\alpha$ is $\{0.05, 0.1, 0.5\}$, and the search space of coefficient of variation $\beta$ is $\{0.05, 0.1\}$. Since PTG is a post-processing algorithm, we do not use any other strategies such as weight averaging (Cha et al., 2021) or ensemble learning (Li et al., 2023), to directly show the influence of PTG.

## F  LEAVE-ONE-DOMAIN-OUT CROSS-VALIDATION RESULTS

We didn't report the results of models selected by leave-one-domain-out cross-validation in the main body, because we recommend that the number of training domains should be 3 at least. In Table 3, we find that even if we use leave-one-domain-out cross-validation, which means we only use two training domains and one validation domain, the performance is still good enough. However, we still suggest that the number of training domains should be adequate just in case.

Table 3: **Leave-one-domain-out cross-validation Results**. All models use ResNet18 as backbones.

| Algorithm | PACS | VLCS | Office-Home | TerraIncognita | Avg |
|---|---|---|---|---|---|
| SelfReg | $83.4 \pm 0.8$ | $78.9 \pm 0.2$ | $66.2 \pm 0.6$ | $46.3 \pm 1.0$ | 68.7 |
| Fish | $84.4 \pm 1.1$ | $80.4 \pm 0.4$ | $65.0 \pm 0.4$ | $43.9 \pm 1.6$ | 68.4 |
| CORAL | $84.7 \pm 0.7$ | $78.9 \pm 0.5$ | $65.9 \pm 0.4$ | $45.8 \pm 1.6$ | 68.8 |
| ERM | $82.7 \pm 1.3$ | $77.0 \pm 0.4$ | $65.5 \pm 1.1$ | $41.2 \pm 0.9$ | 66.6 |
| PTG-Lite | $84.5 \pm 0.3$ | $76.1 \pm 0.2$ | $67.6 \pm 0.2$ | $47.7 \pm 0.7$ | 69.0 |
| ERM-Bayesian | $84.7 \pm 0.5$ | $76.6 \pm 0.4$ | $63.8 \pm 0.4$ | $43.7 \pm 0.9$ | 67.2 |
| PTG | $86.3 \pm 0.4$ | $76.3 \pm 0.5$ | $67.1 \pm 0.3$ | $46.3 \pm 0.7$ | 69.0 |

## G    RESULTS ON RESNET50

We didn't provide results on ResNet50 in the main body for consideration of both GPU memory cost and fairness. We develop a degraded PTG training algorithm to save memory(change the loss of average outputs into average loss of outputs), but this behavior hurts the performance. Consequently, we have reported the performance of all models based on ResNet18 for fair comparison. The performance on ResNet50 is shown in Table 4, where we continue to achieve optimal performance. Again we want to remind that the results on ResNet50 can't reflect the full ability of PTG.

Table 4: **Results on ResNet50**. All methods use training-domain validation set to select models. We report the performance of competitors according to their original papers. We want to remind again that we sacrifice the performance of PTG on ResNet50 to save GPU memory. For more results, please refer to DomainBed.

| Algorithm | PACS | VLCS | Office-Home | TerraIncognita | Avg |
|---|---|---|---|---|---|
| SelfReg | $85.6 \pm 0.4$ | $77.8 \pm 0.9$ | $67.9 \pm 0.7$ | $47.0 \pm 0.3$ | 70.0 |
| Fish | $85.5 \pm 0.3$ | $77.8 \pm 0.3$ | $68.6 \pm 0.4$ | $45.1 \pm 1.3$ | 69.3 |
| CORAL | $86.2 \pm 0.3$ | $78.8 \pm 0.6$ | $68.7 \pm 0.3$ | $47.6 \pm 1.0$ | 70.3 |
| ERM | $85.5 \pm 0.2$ | $77.5 \pm 0.4$ | $66.5 \pm 0.3$ | $46.1 \pm 1.8$ | 68.9 |
| PTG-Lite | $87.3 \pm 0.2$ | $79.6 \pm 0.5$ | $70.0 \pm 0.3$ | $49.2 \pm 0.7$ | 71.5 |
| ERM-Bayesian | $85.8 \pm 0.5$ | $77.7 \pm 0.3$ | $67.1 \pm 0.2$ | $45.5 \pm 0.8$ | 69.0 |
| PTG | $86.7 \pm 0.2$ | $79.4 \pm 0.5$ | $69.4 \pm 0.6$ | $48.5 \pm 1.1$ | 71.0 |

## H    MORE COMBINATIONS

PTG can be combined with most existing methods since it functions as a post-process strategy. However, demonstrating the combination of PTG with all models is unnecessary. By Table 2, we already show that PTG can further promote the performance by combination. We add some experiments to show more combinations in Table 5, which show that all the combinations can bring promotions to the original method.

Table 5: **More Combinations**. All methods use training domain validation to select models. All models use ResNet18 as backbones.

| Algorithm | PACS | VLCS | Office-Home | TerraIncognita | Avg |
|---|---|---|---|---|---|
| ERM | $79.8 \pm 1.2$ | $75.7 \pm 0.2$ | $58.9 \pm 1.0$ | $41.7 \pm 1.5$ | 64.0 |
| PTG | $83.7 \pm 0.1$ | $76.1 \pm 0.5$ | $61.6 \pm 0.4$ | $44.7 \pm 1.2$ | 66.5 |
| SelfReg | $81.8 \pm 1.1$ | $75.3 \pm 1.0$ | $61.2 \pm 0.4$ | $38.2 \pm 2.4$ | 64.1 |
| SelfReg-PTG | $85.3 \pm 0.4$ | $75.2 \pm 0.4$ | $63.6 \pm 0.5$ | $42.6 \pm 0.9$ | 66.7 |
| Fish | $80.7 \pm 0.3$ | $75.9 \pm 0.5$ | $61.2 \pm 0.4$ | $39.0 \pm 1.2$ | 64.2 |
| Fish-PTG | $84.9 \pm 0.2$ | $76.4 \pm 0.3$ | $63.6 \pm 0.4$ | $43.3 \pm 1.1$ | 67.1 |
| CORAL | $81.2 \pm 0.5$ | $75.4 \pm 0.6$ | $61.9 \pm 0.2$ | $38.7 \pm 3.1$ | 64.3 |
| CORAL-PTG | $84.5 \pm 0.4$ | $76.0 \pm 0.8$ | $63.3 \pm 0.8$ | $42.0 \pm 1.3$ | 66.5 |

## I    FULL RESULTS OF TABLE 1

Table 6: **PACS Comparisons**. Out-of-domain classification accuracies(%) on PACS are shown. ERM-Bayesian is a BNN (Blundell et al., 2015) trained by ERM. PTG takes ERM-Bayesian as initialization. PTG-Lite takes ERM as initialization. All models are reproduced on DomainBed. We highlight the **best**, second and t̲h̲i̲r̲d̲ results.

| Algorithm | A | C | P | S | Avg |
|---|---|---|---|---|---|
| CAD | $66.9 \pm 2.7$ | $62.8 \pm 7.9$ | $82.1 \pm 3.7$ | $57.6 \pm 10.8$ | 67.4 |
| IRM | $75.1 \pm 2.5$ | $74.0 \pm 0.9$ | $92.9 \pm 1.6$ | $73.4 \pm 0.6$ | 78.9 |
| MMD | $82.3 \pm 1.4$ | $75.6 \pm 0.9$ | $92.8 \pm 0.2$ | $72.7 \pm 0.5$ | 80.8 |
| ARM | $80.9 \pm 0.6$ | $70.9 \pm 0.5$ | $91.5 \pm 0.3$ | $73.4 \pm 1.2$ | 79.2 |
| GroupDRO | $79.4 \pm 0.9$ | $75.0 \pm 0.6$ | $92.7 \pm 0.3$ | $74.2 \pm 2.0$ | 80.3 |
| VREx | $82.3 \pm 1.6$ | $75.4 \pm 0.6$ | $93.2 \pm 0.5$ | $74.0 \pm 1.7$ | 81.2 |
| Bayes-IRM | $80.9 \pm 0.7$ | $75.5 \pm 1.3$ | $93.7 \pm 0.6$ | $74.2 \pm 1.2$ | 81.1 |
| Mixup | $78.7 \pm 1.8$ | $73.0 \pm 1.2$ | $94.0 \pm 0.3$ | $71.7 \pm 1.0$ | 79.4 |
| Fishr | $84.1 \pm 0.2$ | $74.4 \pm 0.7$ | $92.7 \pm 0.1$ | $73.5 \pm 2.0$ | 81.2 |
| SD | $80.4 \pm 1.3$ | $74.6 \pm 0.5$ | $92.4 \pm 0.2$ | $73.4 \pm 1.2$ | 80.2 |
| SagNet | $79.6 \pm 1.7$ | $75.2 \pm 0.8$ | $93.7 \pm 0.7$ | $76.2 \pm 0.8$ | 81.2 |
| SelfReg | $81.7 \pm 0.8$ | $75.2 \pm 1.3$ | $92.5 \pm 0.4$ | $77.8 \pm 1.1$ | 81.8 |
| Fish | $80.1 \pm 1.2$ | $73.8 \pm 0.8$ | $94.4 \pm 0.2$ | $74.5 \pm 1.0$ | 8̲0̲.̲7̲ |
| CORAL | $79.6 \pm 1.0$ | $75.7 \pm 0.3$ | $94.5 \pm 0.1$ | $75.2 \pm 0.5$ | 81.2 |
| ERM | $79.0 \pm 0.2$ | $74.3 \pm 1.7$ | $94.4 \pm 0.7$ | $71.4 \pm 2.3$ | 79.8 |
| PTG-Lite | $82.4 \pm 0.9$ | $75.0 \pm 0.6$ | $94.9 \pm 0.5$ | $79.6 \pm 0.7$ | 83.0 |
| ERM-Bayesian | $79.2 \pm 1.0$ | $73.9 \pm 0.8$ | $93.6 \pm 0.2$ | $78.6 \pm 0.9$ | 81.3 |
| PTG | $82.6 \pm 0.1$ | $77.0 \pm 0.3$ | $94.7 \pm 0.4$ | $80.6 \pm 0.5$ | **83.7** |

Table 7: **VLCS Comparisons**. Out-of-domain classification accuracies(%) on VLCS are shown. ERM-Bayesian is a BNN (Blundell et al., 2015) trained by ERM. PTG takes ERM-Bayesian as initialization. PTG-Lite takes ERM as initialization. All models are reproduced on DomainBed. We highlight the **best**, second and t̲h̲i̲r̲d̲ results.

| Algorithm | C | L | S | V | Avg |
|---|---|---|---|---|---|
| CAD | $84.9 \pm 5.5$ | $61.3 \pm 0.2$ | $59.0 \pm 3.1$ | $61.3 \pm 0.6$ | 66.6 |
| IRM | $94.7 \pm 1.6$ | $62.6 \pm 0.9$ | $68.6 \pm 1.8$ | $68.7 \pm 4.2$ | 73.6 |
| MMD | $94.4 \pm 1.1$ | $60.7 \pm 2.1$ | $69.5 \pm 1.2$ | $72.0 \pm 4.3$ | 74.2 |
| ARM | $95.4 \pm 1.2$ | $60.3 \pm 1.7$ | $69.0 \pm 2.2$ | $73.4 \pm 1.6$ | 74.3 |
| GroupDRO | $94.5 \pm 1.3$ | $60.6 \pm 1.9$ | $66.7 \pm 1.8$ | $73.9 \pm 1.8$ | 73.9 |
| VREx | $94.5 \pm 1.5$ | $60.5 \pm 2.3$ | $70.2 \pm 1.4$ | $72.3 \pm 2.3$ | 74.4 |
| Bayes-IRM | $94.0 \pm 1.9$ | $62.2 \pm 2.0$ | $69.7 \pm 1.6$ | $72.8 \pm 1.9$ | 74.7 |
| Mixup | $95.5 \pm 0.3$ | $61.0 \pm 0.6$ | $69.2 \pm 1.1$ | $71.7 \pm 1.7$ | 74.4 |
| Fishr | $95.9 \pm 0.9$ | $60.6 \pm 1.5$ | $68.1 \pm 1.2$ | $73.4 \pm 1.7$ | 75.4 |
| SD | $94.8 \pm 0.9$ | $61.3 \pm 1.2$ | $69.2 \pm 0.7$ | $71.6 \pm 1.2$ | 75.0 |
| SagNet | $95.8 \pm 0.9$ | $64.0 \pm 0.8$ | $69.6 \pm 1.0$ | $73.8 \pm 0.9$ | 75.8 |
| SelfReg | $95.4 \pm 0.6$ | $63.2 \pm 1.2$ | $68.9 \pm 1.5$ | $73.4 \pm 0.5$ | 7̲5̲.̲3̲ |
| Fish | $97.0 \pm 0.5$ | $62.3 \pm 1.0$ | $70.7 \pm 0.9$ | $73.5 \pm 0.7$ | 75.9 |
| CORAL | $95.3 \pm 1.2$ | $64.6 \pm 0.9$ | $70.3 \pm 0.7$ | $71.4 \pm 0.2$ | 75.4 |
| ERM | $96.0 \pm 0.3$ | $63.4 \pm 1.1$ | $70.6 \pm 1.2$ | $72.8 \pm 1.2$ | 75.7 |
| PTG-Lite | $96.8 \pm 0.2$ | $63.9 \pm 0.2$ | $69.5 \pm 0.7$ | $72.9 \pm 0.7$ | 75.9 |
| ERM-Bayesian | $96.2 \pm 0.9$ | $62.2 \pm 0.6$ | $67.3 \pm 1.0$ | $70.4 \pm 0.8$ | 74.0 |
| PTG | $97.3 \pm 0.2$ | $64.6 \pm 1.2$ | $68.6 \pm 0.5$ | $73.9 \pm 0.5$ | **76.1** |

Table 8: **OfficeHome Comparisons**. Out-of-domain classification accuracies(%) on OfficeHome are shown. ERM-Bayesian is a BNN (Blundell et al., 2015) trained by ERM. PTG takes ERM-Bayesian as initialization. PTG-Lite takes ERM as initialization. All models are reproduced on DomainBed. We highlight the **best**, second and t̲h̲i̲r̲d̲ results.

| Algorithm | A | C | P | R | Avg |
|---|---|---|---|---|---|
| CAD | $20.9 \pm 6.9$ | $21.3 \pm 9.1$ | $31.4 \pm 11.8$ | $33.0 \pm 11.7$ | 26.6 |
| IRM | $41.3 \pm 5.4$ | $40.4 \pm 2.7$ | $56.6 \pm 5.6$ | $60.4 \pm 5.7$ | 49.7 |
| MMD | $52.5 \pm 0.2$ | $45.3 \pm 0.3$ | $66.3 \pm 0.1$ | $69.5 \pm 0.6$ | 58.4 |
| ARM | $50.8 \pm 0.8$ | $42.9 \pm 0.5$ | $66.0 \pm 0.4$ | $67.2 \pm 0.3$ | 56.7 |
| GroupDRO | $52.4 \pm 0.7$ | $44.7 \pm 1.0$ | $67.0 \pm 0.7$ | $68.0 \pm 0.7$ | 58.0 |
| VREx | $53.4 \pm 0.9$ | $45.7 \pm 0.9$ | $68.0 \pm 0.1$ | $69.6 \pm 0.5$ | 59.1 |
| Bayes-IRM | $51.4 \pm 0.2$ | $46.7 \pm 1.3$ | $70.2 \pm 0.6$ | $68.9 \pm 1.4$ | 59.3 |
| Mixup | $52.0 \pm 1.4$ | $46.9 \pm 0.7$ | $70.2 \pm 0.7$ | $71.0 \pm 0.7$ | 60.0 |
| Fishr | $53.7 \pm 0.5$ | $43.7 \pm 0.4$ | $67.5 \pm 0.5$ | $69.6 \pm 0.1$ | 59.1 |
| SD | $54.4 \pm 1.1$ | $50.1 \pm 0.4$ | $70.3 \pm 0.8$ | $73.8 \pm 0.7$ | **62.2** |
| SagNet | $52.2 \pm 1.4$ | $47.7 \pm 1.4$ | $69.6 \pm 1.1$ | $71.1 \pm 0.8$ | 60.2 |
| SelfReg | $53.0 \pm 1.2$ | $49.2 \pm 0.6$ | $70.2 \pm 0.7$ | $72.4 \pm 0.7$ | 61.2 |
| Fish | $53.9 \pm 0.3$ | $48.8 \pm 1.1$ | $70.0 \pm 0.2$ | $71.9 \pm 0.5$ | 61.2 |
| CORAL | $55.4 \pm 0.9$ | $48.7 \pm 0.2$ | $71.2 \pm 0.6$ | $72.2 \pm 0.3$ | 61.9 |
| ERM | $51.0 \pm 1.6$ | $46.8 \pm 1.4$ | $68.3 \pm 1.2$ | $69.5 \pm 1.5$ | 58.9 |
| PTG-Lite | $53.1 \pm 0.1$ | $48.4 \pm 0.2$ | $70.2 \pm 0.2$ | $72.0 \pm 0.4$ | 60.9 |
| ERM-Bayesian | $51.6 \pm 1.0$ | $48.4 \pm 0.2$ | $66.5 \pm 1.3$ | $70.2 \pm 0.4$ | 59.2 |
| PTG | $55.3 \pm 0.5$ | $50.8 \pm 0.2$ | $69.7 \pm 0.3$ | $70.6 \pm 0.4$ | 6̲1̲.̲6̲ |

Table 9: **TerraIncognita Comparisons**. Out-of-domain classification accuracies(%) on TerraIncognita are shown. ERM-Bayesian is a BNN (Blundell et al., 2015) trained by ERM. PTG takes ERM-Bayesian as initialization. PTG-Lite takes ERM as initialization. All models are reproduced on DomainBed. We highlight the **best**, second and t̲h̲i̲r̲d̲ results.

| Algorithm | L100 | L38 | L43 | L46 | Avg |
|---|---|---|---|---|---|
| CAD | $27.9 \pm 4.7$ | $28.8 \pm 10.7$ | $31.0 \pm 5.2$ | $22.5 \pm 2.8$ | 27.5 |
| IRM | $37.9 \pm 7.6$ | $11.5 \pm 2.4$ | $44.2 \pm 2.9$ | $35.1 \pm 1.2$ | 32.2 |
| MMD | $32.8 \pm 3.0$ | $25.7 \pm 1.0$ | $47.9 \pm 1.9$ | $26.1 \pm 1.8$ | 33.1 |
| ARM | $40.4 \pm 0.7$ | $29.4 \pm 2.4$ | $46.9 \pm 0.8$ | $29.8 \pm 1.3$ | 36.6 |
| GroupDRO | $32.8 \pm 0.7$ | $30.2 \pm 2.1$ | $48.3 \pm 0.9$ | $28.0 \pm 2.1$ | 34.8 |
| VREx | $39.2 \pm 4.3$ | $32.7 \pm 1.3$ | $57.8 \pm 0.8$ | $29.7 \pm 3.1$ | 37.4 |
| Bayes-IRM | $44.0 \pm 2.2$ | $29.8 \pm 3.0$ | $49.6 \pm 0.6$ | $32.0 \pm 2.3$ | 38.9 |
| Mixup | $49.8 \pm 3.6$ | $30.5 \pm 3.9$ | $49.9 \pm 0.8$ | $31.0 \pm 0.8$ | 40.3 |
| Fishr | $42.9 \pm 3.9$ | $36.6 \pm 0.8$ | $48.4 \pm 2.2$ | $32.5 \pm 1.0$ | 40.1 |
| SD | $40.4 \pm 1.8$ | $28.9 \pm 1.7$ | $51.7 \pm 0.6$ | $33.3 \pm 1.2$ | 38.6 |
| SagNet | $42.8 \pm 1.0$ | $27.9 \pm 4.4$ | $51.1 \pm 1.9$ | $35.6 \pm 1.8$ | 39.3 |
| SelfReg | $45.1 \pm 2.0$ | $30.3 \pm 2.1$ | $49.4 \pm 0.4$ | $28.0 \pm 1.7$ | 38.2 |
| Fish | $42.7 \pm 1.4$ | $33.0 \pm 2.9$ | $49.1 \pm 0.6$ | $31.2 \pm 1.4$ | 39.0 |
| CORAL | $45.4 \pm 5.2$ | $27.3 \pm 6.3$ | $51.4 \pm 2.1$ | $30.7 \pm 0.9$ | 38.7 |
| ERM | $49.5 \pm 3.1$ | $32.1 \pm 3.0$ | $50.8 \pm 0.1$ | $34.2 \pm 0.4$ | 41.7 |
| PTG-Lite | $53.1 \pm 1.7$ | $39.2 \pm 0.8$ | $52.1 \pm 0.3$ | $35.1 \pm 0.2$ | **4̲4̲.̲9̲** |
| ERM-Bayesian | $45.3 \pm 3.5$ | $35.3 \pm 1.1$ | $49.7 \pm 1.0$ | $33.5 \pm 1.6$ | 40.9 |
| PTG | $48.6 \pm 0.8$ | $40.7 \pm 0.3$ | $52.7 \pm 0.3$ | $36.8 \pm 0.4$ | 44.7 |

