# OpenReview forum: "Bayesian Domain Invariant Learning via Posterior Generalization of Parameter Distributions"
_ICLR.cc/2024/Conference — Submitted to ICLR 2024_

### Official Review · Reviewer_6EBt · 2023-10-30

**Soundness:** 3 good
**Presentation:** 2 fair
**Contribution:** 3 good
**Rating:** 5
**Confidence:** 3

**Summary:**

This paper proposes to directly learn the domain invariant posterior distribution of network parameters. The theoretical analysis shows that the invariant posterior of parameters can be implicitly inferred by aggregating posteriors on different training domains. Accordingly, this paper proposes a simple yet effective method, named PosTerior Generalization (PTG), that can be used to estimate the invariant parameter distribution.

**Strengths:**

This paper introduces parameter posterior distributions into domain generalization for the first time. And propose two simple yet effective domain generalization methods named Posterior Generalization based on their theories.
The proposed method is simple in theory and algorithm implementation, yet effective as shown by the empirical studies.

**Weaknesses:**

There are some typos, and the writing can be improved.

**Questions:**

1.	Please give clear definitions of $\omega$ and $\theta_i$. How they are related, and how they determine $f_i(\dot)$.
2.	In equations (4), (5) and (6), the notations of expectation and variance do not follow the convention of probability theory. The reviewer fails to figure out what the expectation is taken of, and with respect to what. Even when the reviewer turns to the appendix, it is difficult to understand because $\theta$ and $\omega$ is not clearly defined. The reviewer understands it as the follows: $\omega$ is the model parameter, like the weight of a neural network? $\theta$ corresponds to the mean, $\mu$, and variance, \sigma, when the variational distribution takes Gaussian distribution form.  But if it is the case, by variational inference, \theta is a function of $\omega$ determined by the model (e.g., a neural network); then, how to understand $q(\omega|\theta_i)$. Maybe when your model (the neural network) is not a deterministic model, meaning, both $\omega$ and $\theta$ are random, the above seems right.
3.	The proposed algorithm needs to train a featurizer for each domain. It is computationally expensive.

---

> ### Author Response · Authors · 2023-11-19
> **Overall response**
>
> Thank you for your helpful questions. The discussions will be added to the paper.
>
> Our contributions are: (1) We identified a new research direction to Domain Generalization, the analysis of parameter posterior distributions. (2) We proposed a relaxed assumption and a theory to estimate $p(\omega|\mathcal{D}^c)$ without access to $\mathcal{D}^c$. (3) We proposed two simple methods to approximate $p(\omega|\mathcal{D}^c)$, and the experiments showed the effectiveness. We hope our work can inspire further research.
>
> We hope that these answers can show our ideas more clearly. We are looking forward to your feedback.

---

> > ### Author Response · Authors · 2023-11-21
> > **Summary of repsonse**
> >
> > Thank you for providing suggestions for the paper. The responses may be too long, so we summarize each respnese into one sentence.
> >
> > **question 1**: Sorry this response can't be simplified
> >
> > **question 2**:  $E[p(\omega|\mathcal{D}^c)]$ means $E_{p(\omega|\mathcal{D}^c)}[\omega]$,  $VAR[p(\omega|\mathcal{D}^c)]$ means $VAR_{p(\omega|\mathcal{D}^c)}[\omega]$, $\omega$ is random and $\theta$ (($\mu,\sigma^2$)) is deterministic.
> >
> > **question 3**: The addistional training will not take too much computing resource.
> >
> > We are looking forward to your additional feed back. Please feel free to ask any question, and we will reply immediately.

---

> > > ### Comment · Reviewer_6EBt · 2023-11-22
> > >
> > > Thank the authors for the response. The score remains unchanged.

---

> ### Author Response · Authors · 2023-11-19
> **Question 1**
>
> Thank you for your helpful question. **$\omega$ is the notation of the random-variable-sized parameter**. If we want to refer to posterior parameter distribution, we use $p(\omega|\mathcal{D})$. **$q(\omega|\theta)$ is the variational distribution of parameter, and $\theta$ is the true learnable object**. For example, if we use Gaussian distribution as the variational distribution, then $\theta=(\mu,\sigma^2)$, $\omega$ follows $\mathcal{N}(\mu,\sigma^2)$, and $\mu,\sigma^2$ are the true learnable variable. **We use subscript to denote all the objects that belong to a specific domain**, expect for subscript 0. For example, $p(\omega|\mathcal{D}_i)$ means the posterior parameter distribution given domain $\mathcal{D}_i$. $q(\omega|\theta_i)$ means the variational parameter distribution on $\mathcal{D}_i$. **$f_i()$ is the BNN featurizer on $\mathcal{D}_i$, and its variational parameter distribution is $q(\omega|\theta_i)$**. During training, only $\theta_i$ is updated; during testing, we sample from $q(\omega|\theta_i)$ to form several networks, and take the average of their predictions as the final prediction. Please refer to [1] for more details about BNN. There's no domain $\mathcal{D}_0$, but we keep using $q(\omega|\theta_0)$ and $f_0$ to represent the target parameter distribution and target featurizer.
>
>     [1]Blundell C, Cornebise J, Kavukcuoglu K, et al. Weight uncertainty in neural network[C]//International conference on machine learning. PMLR, 2015: 1613-1622.

---

> ### Author Response · Authors · 2023-11-19
> **Question 2**
>
> Thank you for your helpful question. We will correct the notations. $E[q(\omega|\theta_i)]$ means $E_{q(\omega|\theta_i)}[\omega]$, and $VAR[q(\omega|\theta_i)]$ means $VAR_{q(\omega|\theta_i)}[\omega]$. If we use Gaussian distribution as the variational distribution on each domain, then $E[q(\omega|\theta_i)]=\mu_i$,$VAR[q(\omega|\theta_i)]=\sigma^2_i$. However, if the variational distribution is other distributions, Equation 4,5 still hold. $q(\omega|\theta_i)$ means the variational distribution, where $\theta_i$ is the parameter of this distribution. For Gaussian variational inference, $\theta_i=(\mu_i,\sigma^2_i)$ and $q(\omega|\theta_i) = \mathcal{N}(\mu_i,\sigma^2_i)$ (or $\mathcal{N}(\omega;\mu_i,\sigma^2_i)$ for better understanding). $\theta_i$ is deterministic, $\omega$ is random.

---

> ### Author Response · Authors · 2023-11-19
> **Question 3**
>
> Thank you for your helpful question. The computational costs are close to ERM, given that **PTG only need a few more iterations** (50 iterations, about 1.4 epochs) to aggregate the learned posteriors. **Despite training three models, each model is trained on one domain representing 1/3 of the entire training data, so the overall computationally expensive is still same as ERM**. The aggregation computations also have close forms. The training duration for ERM-Bayesian is approximately 44 minutes, followed by PTG training which spans approximately 20 minutes.
>
> However, the GPU memory cost is expensive because PTG has to load the parameters of all models for training and computing the aggregated posteriors. We have addressed this phenomenon in Section 5. Simplified structure, parallel computation or simply making aggregations by CPU may be solutions, but they are future works.

---

### Official Review · Reviewer_EkMw · 2023-10-30

**Soundness:** 3 good
**Presentation:** 2 fair
**Contribution:** 2 fair
**Rating:** 6
**Confidence:** 4

**Summary:**

The paper proposes a posterior generalization method for domain invariant learning. Different from the previous methods that do invariant learning on features, the proposed method directly infers the invariant posterior of the parameters by posterior aggregation. The authors also propose a lite version of posterior generalization for widespread applications. Experiments on DomainNet show the effectiveness of the proposed method.

**Strengths:**

1. The idea of directly inferring the invariant posterior of the model parameters is novel and interesting.

2. The experiments show the effectiveness of the proposed method.

**Weaknesses:**

1. Although the idea is interesting, it is still unclear why directly inferring the domain invariant model posteriors is better than feature invariant learning. The authors argue that the proposed method can extract more invariant information from features but the reason is also not clear.

2. As the author posted, the proposed method is not memory efficient since BNNs require more parameters than the deterministic models.

**Questions:**

It seems there are three stages of PTG, first pre-training the BNN models by ERM, then learning source domain-specific posteriors, at last generating the invariant posterior and training the classifier. It is better to make this procedure clear in the paper for easier following. And there are some details also need to be clarified.

1. Before pretrained on the source domains by ERM, is the BNN parameters initialized randomly or by ImageNet pretrained parameters?

2. In eq.(4), what are the mean and variance for each BNN convolutional layer? are they channel-wise vectors or scalers?

3. once obtained the mean and variance of the invariant posterior f_0, how to sample the parameters? By Monte-Carlo sampling?

4. In the algorithm, I found that in each training iteration, the source-specific posterior f_i is updated first, then the method aggregates them as the invariant posteriors and trains it together with the classifiers. However, in the next iterations, the f_i is updated again without considering any information in f_0, and f_0 is updated according to the new f_i, also without the f_0 in the previous iterations. If it is, why need to update f_0 by the cross-entropy loss functions?

---

> ### Author Response · Authors · 2023-11-19
> **Overall response**
>
> Thank you for your helpful questions. The discussions will be added to the paper.
>
> Our contributions are: (1) We identified a new research direction to Domain Generalization, the analysis of parameter posterior distributions. (2) We proposed a relaxed assumption and a theory to estimate $p(\omega|\mathcal{D}^c)$ without access to $\mathcal{D}^c$. (3) We proposed two simple methods to approximate $p(\omega|\mathcal{D}^c)$, and the experiments showed the effectiveness.
>
> We hope our work can inspire further research. We hope that these answers can show our ideas more clearly. We are looking forward to your feedback.

---

> > ### Author Response · Authors · 2023-11-21
> > **Summary of response**
> >
> > Thank you for providing suggestions for the paper. The responses may be too long, so we summarize each respnese into one sentence.
> >
> > **weakness 1**: Our aim is to learn $p(\omega|\mathcal{D}^c)$, which means the posterior parameter distributions that use all domain invariant information for prediction
> >
> > **weakness 2**:There are works that improve BNN memory efficiency, but our focus is on Domain Generalization.
> >
> > **question 1**: We initialize a BNN by a ResNet pretrained on ImageNet.
> >
> > **question 2**:  We use a diagonal Gaussian variational distribution to approximate the distribution of the whole parameters, and $(\mu,\sigma^2)$ means the expectation and covariance.
> >
> > **question 3**: We use Monte-Carlo sampling to sample from variational distributions.
> >
> > **question 4**:  Both algorithmically required and to prevent potential problems in application.
> >
> > We are looking forward to your additional feed back. Please feel free to ask any question, and we will reply immediately.

---

> > > ### Comment · Reviewer_EkMw · 2023-11-22
> > > **Thanks for the authors' response**
> > >
> > > The response of the authors solves most of my concerns. I would increase the score to 6.

---

> ### Author Response · Authors · 2023-11-19
> **Weakness 1**
>
> Thank you for your insightful question. From Bayesian view, **our aim is to learn $p(\omega|\mathcal{D}^c)$, which means the posterior parameter distributions that use all domain invariant information for prediction.** There may be a misunderstanding that by "domain invariant posterior", we refer to $p(\omega|\mathcal{D}^c)$ rather than parameters that vary little among training domains. We made this description in Section 1 paragraph 4, and we will improve our expression for better clarity. We explain in Section 3.5 about the relationship between domain invariant parameters $p(\omega|\mathcal{D}^c)$ and its variation rate.
>
> We also explain why **domain specific features may still contain domain invariant information** in Section 3.5. In Fig 2, there are stripes on feature maps. These strips are domain specific features because one is straight and one is wavy. If we use MSE to evaluate the distance between these features, they are far away (only few points overlap). We show by this toy model that this information can be extracted by parameters, but if we only want to extract domain invariant features, this information will be ignored. Fig 2 is a toy model that, for easy understanding, we assume the domain information exists in the feature map. However, we don't know exactly how the information is represented or how to get it, and we don't care. We only assume that $\mathcal{D}^c$ and $\mathcal{D}^v$ exist and are independent, and the domain distribution $p(\mathcal{D})$ can somehow be represented by them. One of our contributions is that we can estimate $p(\omega|\mathcal{D}^c)$ without access to $\mathcal{D}^c$.

---

> ### Author Response · Authors · 2023-11-19
> **Weakness 2**
>
> Thank you for your helpful question. **We didn't improve BNN memory efficiency because our focus was on Domain Generalization rather than Bayesian deep learning.** For easy understanding, we only provide PTG based on primary variational inference BNN [1]. There are many further works that focus on reducing the memory cost, such as [2,3]. As a result, there may be more solutions that can reduce the memory cost of PTG, but it's beyond this paper and could be future works. Our main concern is how to solve the problem of DG from the perspective of parameter posterior, and whether our solution is effective. We theoretically show how to estimate the posterior given invariant information by posteriors on each training domain. We further explain why the aggregated posterior can learn domain invariant information from the view of representation learning. A practical implementation is presented method based on VI, and the effectiveness is validated by experiments.
>
>     [1]Blundell C, Cornebise J, Kavukcuoglu K, et al. Weight uncertainty in neural network[C]//International conference on machine learning. PMLR, 2015: 1613-1622.
>     [2]Kristiadi A, Hein M, Hennig P. Being bayesian, even just a bit, fixes overconfidence in relu networks[C]//International conference on machine learning. PMLR, 2020: 5436-5446.
>     [3]Deng Z, Zhou F, Zhu J. Accelerated Linearized Laplace Approximation for Bayesian Deep Learning[J]. Advances in Neural Information Processing Systems, 2022, 35: 2695-2708.

---

> ### Author Response · Authors · 2023-11-19
> **Question 1**
>
> Thank you for your helpful question. We initialize a BNN by a ResNet pretrained on ImageNet, which is the common choice in Domain Generalization. The method to initialize a BNN by DNN is adopted from [1]
>
>     [1]Krishnan R, Subedar M, Tickoo O. Specifying weight priors in bayesian deep neural networks with empirical bayes[C]//Proceedings of the AAAI Conference on Artificial Intelligence. 2020, 34(04): 4477-4484.

---

> ### Author Response · Authors · 2023-11-19
> **Question 2**
>
> Thank you for your careful reading and question. Equation 4 is a common, simplified notation in variational inference BNN[1]. **$\omega$ represents a vector of the whole parameters**. We flatten all the parameters in a BNN and concatenate them to form $\omega$(just in concept, not in practice). Usually, **we use a diagonal Gaussian variational distribution to approximate $\omega$**. As a result, **we can represent the mean and covariance of the variational distribution by vector $\mu$ and $\sigma^2$**. In addition, BNN can still be trained by backward propagation, which means different parts of $\omega$ can be trained individually. For example, if we want to train a BNN convolutional layer, there are in fact two trainable convolutional layers: one represents $\mu$ and one represents $\sigma$. They have the same structure as the intended BNN convolutional layer. Please refer to [1] for more BNN training details.
>
>     [1]Blundell C, Cornebise J, Kavukcuoglu K, et al. Weight uncertainty in neural network[C]//International conference on machine learning. PMLR, 2015: 1613-1622.

---

> ### Author Response · Authors · 2023-11-19
> **Question 3**
>
> Thank you for your careful reading and question. We use Monte-Carlo sampling to sample from variational distributions. This is a common operation in variational inference BNN.[1]
>
>     [1]Blundell C, Cornebise J, Kavukcuoglu K, et al. Weight uncertainty in neural network[C]//International conference on machine learning. PMLR, 2015: 1613-1622.

---

> ### Author Response · Authors · 2023-11-19
> **Question 4**
>
> Thank you for your insightful question. First, we didn't update classifier on each domain, and after the change of $f_0$, we have to update classifier to adapt it to $f_0$. Second, in theory PTG don't need to update $f_0$ after the aggregation. However, we are afraid that after some iterations, $f_i$ may get diversified too much that most posteriors in $f_0$ may have too large variance. We are not sure that whether a good generalization ability can be achieved in this stage, but too large variance may make a BNN divergent. As a result, we further update $f_0$ by just one iteration, so $f_0$ will not change a lot, and it can still extract meaningful features.

---

### Official Review · Reviewer_mZKN · 2023-10-30

**Soundness:** 3 good
**Presentation:** 1 poor
**Contribution:** 1 poor
**Rating:** 6
**Confidence:** 4

**Summary:**

The work focuses on tackling domain generalization from the view of parameter posterior learning, aiming to extract invariant “information” for better generalization across training domains. Their work is built upin Bayesian Neural Networks (BNNs), the approach directly learns domain invariant posterior distributions of network parameters. A theorem (conditional distribution marginalization) shows implicit learning of invariant posteriors by aggregating network parameter posteriors from different domains, allowing for a relaxed assumption. The proposed PosTerior Generalization (PTG) employs variational inference to estimate the invariant parameter distribution, demonstrating competitive performance on diverse benchmarks. The proposed PTG is built upon the existing DG methods, where PTG needs the existing DG methods to provide initilization for their parameter posterior aggregation.

**Strengths:**

1. The explanation of the intractability of the defined problem is clear, offering a compelling rationale for employing variational inference to estimate the posterior density.

2. The paper is well-written with a clear flow to understand the proposed PTG.

**Weaknesses:**

## Majors:

1. The lack of definitions for "parameter" and "feature" leaves me puzzled about the proposed method. What constitutes the samples of the posterior after the training of the Bayesian neural net converges—model parameters, latent features, or something else entirely? Furthermore, clarity is needed on the dimensionality of the samples from the proposed posterior.

2. There is no need to present p(w|D^{c}) = E_{p(D^{v})}(p(w|D^{c}, D^{v})) as a theorem; it is a standard statistical technique known as marginalization. Routine operations in statistics, like this, do not necessitate proof.

3. I could not identify any specific contributions made by the authors in this paper. It reads more like a review of Bayesian methods in domain generalization. The claimed theorem appears to be about the marginalization of conditional distributions. I am interested in the authors' elaboration on this during the rebuttal.

4. I'm sensing a concern about the assumption that \mathcal{D}_{c} ​ remains constant across the entire dataset \mathcal{D}, especially given the notation \mathcal{D}_{i}^{N} ​ mentioned by the authors. It seems the authors are questioning whether there can be a subset of data within one dataset that is fully domain-specific or domain-invariant, as opposed to only certain features (latent encoded outputs) being domain-invariant. Consequently, I find the motivation behind the proposed work to be lacking. Let us break it down. The authors seem to suggest that \mathcal{D}_{c} remains consistent throughout \mathcal{D}, where \mathcal{D}_{i}^{N} ​ represents subsets of data within the dataset. The authors point out that there is a subset of samples that are domain-specific or invariant rather than there is part of feature elements within entire data samples that are domain-specific or invariant (e.g., In the conventional understanding, we acknowledge that images of dogs often exhibit diverse background information, which can be seen as domain-specific. Simultaneously, the outlines or features of dogs themselves are considered domain-invariant. However, the authors introduce the idea that within the broader category of dogs, there should be a subset of images that are specifically tied to a domain or completely domain-invariant. This assertion does not seem to align with the typical variability observed in dog images, and I find it challenging to envision such a subset existing within this category). This leads to a potential misalignment in the assumption. If the proposed work relies on the idea that certain data samples are entirely domain-specific within a dataset, it might not be well-grounded. It might be more accurate to frame the motivation around features being domain-invariant rather than assuming entire data samples share this characteristic. In essence, it appears there's room for clarification or adjustment in the assumption to better reflect the reality of domain invariance within datasets. This refinement would strengthen the motivation behind the proposed approach.

5. The literature compared in this study appears to be somewhat outdated, with the highlighted method, SOTA CORAL, dating back to 2017. Given that the proposed method relies on an existing domain generalization (DG) method for initialization, it becomes crucial to benchmark against more recent approaches. It would be particularly insightful to demonstrate how the proposed method not only aligns with but also enhances the state-of-the-art frameworks in domain generalization. Please consider conducting comparisons with the latest methods to provide a more comprehensive evaluation of your proposed approach in the context of contemporary DG frameworks.

## Minors:

1. The authors should perform an ablation study to assess the performance of the proposed PTG without relying on other domain generalization (DG) methods for initialization. For instance, evaluating PTG using only the pre-trained ResNet-18 as the initialization for model training on each domain would provide insights into its standalone effectiveness.

2. Correct the grammatical errors, for example, on page 4, section 3.2, the proof should be "shown" instead of "show."

### Reference

[1] Zhang, X., He, Y., Xu, R., Yu, H., Shen, Z. and Cui, P., 2023. Nico++: Towards better benchmarking for domain generalization. In Proceedings of the IEEE/CVF Conference on Computer Vision and Pattern Recognition (CVPR) (pp. 16036-16047).

**Questions:**

1. The authors in section 3.3 suggest initializing the BNN on each domain with a uniformly well-generalized model, such as a BNN trained using Empirical Risk Minimization (ERM). The confusion arises regarding the connection between initializing the model on each domain and selecting the best model through ERM. To clarify, the authors advocate training a BNN using ERM on a specific dataset before initializing it on different domains. The dataset used for this ERM training is not explicitly mentioned in this section, but it is the dataset that provides the well-generalized model used as the starting point for initialization across various domains. In essence, the ERM-trained BNN serves as a kind of starting point. It is trained on a particular dataset that captures general patterns. My understanding is that this well-generalized model is then employed as the initial configuration for the BNN when dealing with different domains. The emphasis is on using a model that has already demonstrated good generalization capabilities through ERM, ensuring a solid starting point for adaptation to diverse domains.

2. The terminologies used in the paper are confusing. In the introduction, the authors equate domain-invariant features to domain-invariant parameters. In typical deep-learning contexts, parameters refer to learnable model weights and biases, while features denote the encoder's output. To prevent potential misinterpretations, it would be helpful if the authors could offer clear definitions for parameters and features within the scope of their work.

3. Could you provide clarity on the definition of the prior model in the proposed work? Is it initialized from the pretraining of ResNet, or is it initialized from the pretraining of another domain generalization (DG) method? Please elaborate on this aspect.

4. Could you specify the space in which domain-invariant and domain-specific information is defined? Is it in the input space, latent space, or output probability space? If it's in the input space, how does \mathcal{D} change? If it's in the latent and output spaces, how does p(\mathcal{D}^{c}) remain constant? Clarification on this aspect would enhance understanding.

5. In Algorithm 1, when the authors mention updating the model parameters with a Gaussian distribution, are they referring to sampling the model parameters from the Gaussian distribution outlined in Equation (4)? If so, could you provide clarification on the method used for this sampling process? Is it achieved through random sampling? Please provide additional details.

6. In Figure 2, the authors mention extracting domain-invariant information from domain-specific features (Z^V). Given that these features reside in the encoded space (the output space of ResNet-18) if the training of ResNet-18 is indeed effective in extracting domain-invariant features, it raises a question: How can domain-invariant information also be present within features specifically identified as domain-specific? The term "information" is used without a clear definition from the authors. This ambiguity, coupled with an unclear definition of \mathcal{D}, restricts readers from fully grasping the paper's content. Could the authors provide more clarity on interpreting "information" and a more explicit definition of \mathcal{D}? My understanding is that the authors are trying to claim that the feature map might not effectively capture the domain-invariant information from the data. If my understanding is correct, then the claim is very ambiguous. The authors should at least provide some ways to quantify or visualize it. In the current version of the paper, this claim is not supported theoretically or empirically.

---

> ### Author Response · Authors · 2023-11-19
> **Overall response**
>
> Thank you for your helpful questions. The discussions will be added to the paper. We hope that these answers can express our ideas more clearly. We are looking forward to your feedback.

---

> ### Author Response · Authors · 2023-11-19
> **Weakness 1**
>
> We are sorry that we made the confusion. In Bayesian Neural Networks, posterior usually refers to the **posterior distribution of parameters given data $p(\omega|D)$**. We didn't make new definitions for parameter and feature. Parameter means model parameters and feature means latent features. We sample for 5 times from the variational distribution $q(\omega|\theta)$, and $q(\omega|\theta)$ is the approximation of the true parameter posterior $p(\omega|D)$. The dimensionality of posterior is the dimensionality of parameter. For example, the posterior of a 3X3X3 conv layer in a BNN has the dimensionality 3X3X3. The variational distribution $q(\omega|\theta)$ also share the same dimensionality as $p(\omega|D)$.  The principle of BNN and the procedure of variational inference is shown in Section 2.3, 2.4 and 3.1, and we will add more descriptions.

---

> ### Author Response · Authors · 2023-11-19
> **Weakness 2**
>
> Thank you for your helpful suggestion. **Maybe there is a misunderstanding between $\int P(X|Y=y)P(Y=y)dy$ and $\int P(X|Y=y,Z)P(Y=y)dy$. **Marginalization usually means the process that requires summing over the possible values of one variable to determine the marginal contribution of another, to be specific, $P(X)=\int P(X,Y=y)dy=\int P(X|Y=y)P(Y=y)dy$. However, **our theorem, $p(\omega|\mathcal{D}^c)=\int p(\mathcal{D}^v)p(\omega|\mathcal{D}^c,\mathcal{D}^v)d\mathcal{D}^v$, follows another Bayesian principle.** The proof of Theorem 3.1 is shown in appendix B. Although $p(\omega|\mathcal{D}^c)=\mathbb{E}_{p(\mathcal{D}^v)}[p(\omega|\mathcal{D}^c,\mathcal{D}^v)]$ appears to be true, it only holds under the condition that $\mathcal{D}^c$ and $\mathcal{D}^v$ are independent. Besides, Theorem 3.1 shows that we can estimate $p(\omega|\mathcal{D}^c)$ by the posterior given full information, which plays a vital role in the following context.

---

> ### Author Response · Authors · 2023-11-19
> **Weakness 3**
>
> Thank you for your insightful question. One of our contributions is that **we are the first one to align parameter posteriors for domain generalization.** As introduced in Section 1 paragraph 3 and Section 2.3, all former works use BNN to show the distribution of latent features, and they align the feature distributions similar as other Non-Bayesian ways. We find that the alignment of parameter distributions can also bring improvements. Besides, **we propose a method to estimate the posterior given domain invariant information $p(\omega|\mathcal{D}^c)$,** which can achieve the best domain generalization ability in theory. There's a problem that $\mathcal{D}^c$ is unknown, but we can still estimate $p(\omega|\mathcal{D}^c)$ without access to $\mathcal{D}^c$. In addition, although we propose the way to estimate $p(\omega|\mathcal{D}^c)$, **we solve some difficulties in implementation**, such as disordered dimensions of parameters. We explained these difficulties in section 3.3 and 3.4. We propose two simple yet effective implementations to solve these problems, and experiments show their superiority. In conclusion, **we identify a new research direction to Domain Generalization**, the analysis of parameter posterior distributions, and we propose the basic framework for it.

---

> ### Author Response · Authors · 2023-11-19
> **Weakness 4**
>
> Thank you for your insightful questions. We are sorry that we caused some misunderstanding.
>
> **The key misunderstanding is, $\mathcal{D}^c$ and $\mathcal{D}^v$ are not dataset! The second misunderstanding is, it is $p(\mathcal{D}^c)$ that remains constant, rather than $\mathcal{D}^c$.**
>
> **$\mathcal{D}^c$ and $\mathcal{D}^v$ are two abstract concepts**: $\mathcal{D}^c$ means the collection of all domain invariant information from a domain $\mathcal{D}$, and $\mathcal{D}^v$ is the opposite. $\mathcal{D}$ refers to a domain rather than a dataset, $p(\mathcal{D}^v)$ is its distribution, and the corresponding dataset are samples from $p(\mathcal{D}^v)$. We use $\mathcal{D}_i$ to distinguish between different domains. We assume that $\mathcal{D}^c$ and $\mathcal{D}^v$ are two independent sufficient statistics of $\mathcal{D}$. To be specific, there exist a bijection between $p(\mathcal{D})$ and $p(\mathcal{D}^c)p(\mathcal{D}^v)$, and $p(\mathcal{D}^c)$ and $p(\mathcal{D}^v)$ can be inferred by the dataset of $\mathcal{D}$. However, we don't care about (1) the specific algebra expression of $\mathcal{D}^c$, (2) the specific bijection formula, (3) the specific formula to estimate $p(\mathcal{D}^c)$ from dataset. One of our contributions is to estimate $p(\omega|\mathcal{D}^c)$ without access to $\mathcal{D}^c$.
>
> In your example, $\mathcal{D}$ means a domain of dogs, and $\mathcal{D}_i$ may refer to a specific kind of dog, like golden retriever. $\mathcal{D}^c$ means the common information that all dogs share, such as the overall body shape, the facial patterns, the teeth and so on. $\mathcal{D}^v_i$ means the specific information that golden retriever have, such as the color, the hair length. As stated before, we don't care how the information is represented or how to get it, we just assume that it exists. We can directly estimate the posterior parameter distributions that use the common information for recognition. This is why we claim that our assumption is more relaxed than domain invariant features.
>
> We explained our idea in Section 1 paragraph 5, Section 3.1 and Appendix A. Thanks again for your helpful question, we will add these discussions and your example in the paper.

---

> ### Author Response · Authors · 2023-11-19
> **Weakness 5**
>
> Thank you for your valuable suggestions. Our comparative analysis remains contemporary considering that more than half of the models(8/14) are introduced after 2021, and the rest are classical counterparts. We selected our competitors for a fair comparison. Model selection plays a vital role in domain generalization, so **we only compare with models available on DomainBed**, which is a fair test platform that automatically select models. **We did not compare PTG with many other SOTA methods, because they either use CLIP pretrained model, SWAD weight averaging strategy, large-scale ensemble learning or other strategies.**
>
> We are sorry that we can't combine PTG with EoA, which is the best model in the benchmark NICO++. Because EoA is an ensemble model, while PTG requires a single model as its prior. However, we want to mention that although CORAL is old, it still gets the third performance in NICO++ and there is only a small distance.(EoA 78.88, MIRO 78.65, CORAL 78.38). As you suggested, we provide the result of the best two models on NICO++, but please note we still give results on DomainBed, and all models use ResNet50.  The results of MIRO and EoA are collected form [1]. Please refer to Section 4.1 paragraph 3 and Appendix G for more details.
>
> | Algorithm | PACS | VLCS | Office-Home | TerraIncognita | Avg |
> | :-------: | :--: | :--: | :---------: | :------------: | :--: |
> |    PTG    | 86.7 | 79.4 |    69.4    |      48.5      | 71.0 |
> |   CORAL   | 86.2 | 78.8 |    68.7    |      47.6      | 70.3 |
> |   MIRO   | 85.4 | 79.0 |    70.5    |      50.4      | 71.3 |
> |    EoA    | 88.6 | 79.1 |    72.5    |      52.3      | 73.1 |
>
> We admit that EoA is better than PTG, but we want to remind again that EoA is an ensemble method while PTG just has one model in the testing stage. We already provide more combinations in Appendix H, and we are sorry that we can't provide the combination with MIRO due to the time limit.
>
>     [1] Arpit D, Wang H, Zhou Y, et al. Ensemble of averages: Improving model selection and boosting performance in domain generalization[J]. Advances in Neural Information Processing Systems, 2022, 35: 8265-8277.

---

> ### Author Response · Authors · 2023-11-19
> **Minors**
>
> Thank you for your careful reading. We will modify these problems. The result of PTG on ResNet 18 without prior network is:
>
> | Prior | PACS | VLCS | Office-Home | TerraIncognita | Avg |
> | :---: | :--: | :--: | :---------: | :------------: | :--: |
> |  w/  | 83.7 | 76.1 |    61.6    |      44.7      | 66.5 |
> |  w/o  | 79.9 | 74.5 |    56.3    |      38.2      | 62.2 |

---

> ### Author Response · Authors · 2023-11-19
> **Question 1**
>
> Thank you for your insightful question. **We train the initial network by ERM just on all the given training domains. There's no additional training dataset.** The ERM initialization process is part of our method, and since ERM introduces no generalization techniques, we think the improvements comes from our work.

---

> ### Author Response · Authors · 2023-11-19
> **Question 2**
>
> We are sorry that we caused this confusion. Feature and parameters are just common defined. The domain invariant features are also same as common works. **Domain invariant parameters (domain invariant posteriors) denote the posterior distribution of parameters given domain invariant information $p(\omega|\mathcal{D}^c)$.** We give the definitions in Section 1 paragraph 4 and section 3.1, and we will add the definition by definition formula.

---

> ### Author Response · Authors · 2023-11-19
> **Question 3**
>
> Thank you for your helpful question. **There are two initialization procedure.** First, we initialize a BNN by a ResNet pretrained on ImageNet, which is the common choice in Domain Generalization. The method to initialize a BNN by DNN is adopted from [1]. Second, after the ERM training, we initialize the BNN on each domain by the ERM trained BNN. We are sorry that we didn't draw a clear distinction between these prior models. We will modify the expression.-
>
>     [1]Specifying weight priors\mu in bayesian deep neural networks with empirical bayes.

---

> ### Author Response · Authors · 2023-11-19
> **Question 4**
>
> Thank you for your helpful question. As stated in Weakness 4, we don't have to specify these aspects. One of our contributions is to estimate $p(\omega|\mathcal{D}^c)$ without access to $\mathcal{D}^c$.

---

> ### Author Response · Authors · 2023-11-19
> **Question 5**
>
> There seems to by a misunderstanding of BNN. $\omega$ usually means the learnable parameters in DNN. However, in variational BNN, $\omega$ is the notation of the random-variable-sized parameter, $q(\omega|\theta)$ is the variational distribution of parameter, and $\theta$ is the true learnable object. For example, if we use Gaussian distribution as the variational distribution, then $\omega$ follows $\mathcal{N}(\mu,\sigma^2)$, and $\mu,\sigma^2$ are the true learnable variable. During training, we only learn $\mu,\sigma^2$ by Equation 4 and 5. During testing, we sample $\omega$ from $\mathcal{N}(\mu,\sigma^2)$ for 5 times to form 5 different models, and use the average of their predictions as the final prediction.

---

> ### Author Response · Authors · 2023-11-19
> **Question 6**
>
> Thank you for your insightful question. In Fig 2, there are stripes on feature maps. These strips are domain specific features because one is straight and one is wavy. If we use MSE to evaluate the distance between these features, they are far away (only few points overlap). We show by this toy model that this information can be extracted by parameters, but if we only want to extract domain invariant features, this information will be ignored. Fig 2 is a toy model that, for easy understanding, we assume the domain information exists in the feature map. However, we don't know exactly how the information is represented or how to get it, and we don't care. We only assume that $\mathcal{D}^c$ and $\mathcal{D}^v$ exist and are independent, and the domain distribution $p(\mathcal{D})$ can somehow be represented by them. One of our contributions is that we can estimate $p(\omega|\mathcal{D}^c)$ without access to $\mathcal{D}^c$.

---

> ### Author Response · Authors · 2023-11-21
> **Summary of response**
>
> Thank you for providing suggestions for the paper. The responses may be too long, so we summarize each respnese into one sentence.
>
> **weakness 1**:  Parameter means model parameters, feature means latent features and during testing we sample from parameter distribution.
>
> **weakness 2**: Theorem 3.1 is not marginalization, and it plays a vital role in the following context.
>
> **weakness 3**: We are the first one to align parameter posteriors for domain generalization, exploring a new research direction.
>
> **weakness 4**: $\mathcal{D}^c$ and $\mathcal{D}^v$ are not datasets.
>
> **weakness 5**: Our experiment remains contemporary, and we add some suggested competitors.
>
> **mino** : we add the suggested ablation study, and we will fix typos.
>
> **question 1**: ERM is done on all the given training domains. There's no additional training dataset.
>
> **question 2**: Feature and parameters are just common defined. Domain invariant parameters means $p(\omega|\mathcal{D}^c)$.
>
> **question 3**: Both two initialization procedures exist.
>
> **question 4**: We don't have to specify these aspects to build PTG.
>
> **question 5/6**: Sorry the responses can't be simplified
>
> We are looking forward to your additional feed back. Please feel free to ask any question, and we will reply immediately.

---

> > ### Comment · Reviewer_mZKN · 2023-11-22
> > **Reply to The Author's Response**
> >
> > Thank you for providing detailed response to address my concerns.
> >
> > [**W1**] well-addressed by the authors.
> >
> > [**W2**] well-addressed by the authors.
> >
> > [**W3**] I am a bit surprised that domain generalization people haven't delved into aligning the posterior distributions of model parameters. From my understanding, the primary goal of BNN is to infer the posterior distribution for model parameters rather than the feature distribution.
> >
> > [**W4**] well-addressed by the authors.
> >
> > [**W5**] I might need more time to think this.
> >
> > [**minor**] I am wondering if all the baseline methods use the prior work as initialization? If the initialization settings are different, then how could the authors ensure the comparisons are fair? Now, it becomes my major concern.
> >
> > [**Q1**] well-addressed by the authors.
> >
> > [**Q2**] well-addressed by the authors.
> >
> > [**Q3**] If ERM initialization is one contribution as claimed by the authors. I am wondering how the baseline methods will perform when they are initialized with the same initialization? Control variables should be used for all experiments.
> >
> > [**Q4**] well-addressed by the authors.
> >
> > [**Q5**] well-addressed by the authors.
> >
> > [**Q6**] the authors' focus is not the alignment from data or feature level, but the model parameters. Then, there is too much discussion given with respect to the domain invariant information. With very ambugious definitions on the domain invariant information, it makes the readers hard to grasp the authors' main idea.
> >
> > I would appreciate it if the authors could provide the revised PDF with the changes highlighted in blue. The focus should be on reducing contextual information related to domain-invariant aspects, while focusing on the core idea of the paper—model parameter alignment. I would like to review this before the discussion deadline, as it may influence my decision to consider a higher score.

---

> > > ### Author Response · Authors · 2023-11-23
> > > **New reply**
> > >
> > > Thank you for your detailed reply. The current manuscript has reached the maximum page capacity, and we have to make major modification to add the discussions into the paper. We are working hard, but there's very little time left. We will upload the PDF once finished and possible. No matter the outcome, we appreciate your dedication.
> > >
> > > ## Minor
> > >
> > > PTG use ERM-trained network as its prior, and other competitors are initialized by ImageNet-pretrained network. **The ERM training or other training procedure on training domains should be considered as part of our algorithm.** As explained in section 3.3, PTG requires the same initialization for individual networks, and the initialization should fit the training domains well. Besides, we use ERM as the initialization, which means no additional DG strategy are included to PTG.
> > >
> > > Second, **additional training do not mean better generalization**. The performance on training domains can become better as the epochs increase, but it may cause the problem of over-fitting that hurts the performance on differently distributed test domains. Specifically, DomainBed has to store copies during the whole training process, just to select the best early-stopped model instead of the final model. As a result, even if we add additional training besides ERM, it is our method that works, otherwise the additional training will lead to worse performance on test domains.
> > >
> > > Third, **different initialization setting is a common choice in Domain Generalization**. For example, MIRO[1] use CLIP pretrained network as its initialization. As stated in the second answer, simply use a different pretrained model may not bring significant improvement. It's the way to use pretrained model matters, rather than the use of pretrained model alone.
> > >
> > >     [1]Cha J, Lee K, Park S, et al. Domain generalization by mutual-information regularization with pre-trained models[C]//European Conference on Computer Vision. Cham: Springer Nature Switzerland, 2022: 440-457.
> > >
> > > ## Q3
> > >
> > > Thank you for your suggestion. We didn't initialize other models by ERM, because we don't know whether this behavior helps or hurts these DG methods. A possible situation is, a DG method may stay in an extreme point that far away form ERM, and the ERM fine-tune(compared with ImageNet pretraining) may make the model converge to an unexpected extreme point. However, we appreciate this suggestion, and we will add the suggested experiment in our ablation study.
> > >
> > > ## Q6
> > >
> > > Thank you for your help suggestion. We want to estimate $p(\omega|\mathcal{D}^c)$, so we have to first point out what $\mathcal{D}^c$ means. We will change the description into a more brief and formal way.

---

> ### Comment · Reviewer_mZKN · 2023-11-23
> **Reply to The Authors**
>
> Thanks so much to the authors for the further detailed elaborations and significant effort during the author response period. The authors have now addressed most of my concerns. **For now, I am happy to raise my score to 6**. Meanwhile, I would still like to see a fairer comparison when the baseline methods are empowered with ERM initialization (to have the same initialization as the proposed method). However, as indicated by the authors, ERM initialization is also a part of their contribution. I would like to suggest the authors include it in the ablation studies.
>
> I might need more time to think about it and re-evaluate the paper again. If the authors agree to provide the ablation studies (including comparison with the baselines empowered with ERM initialization) in the camera-ready version and my re-evaluation ends up telling me that it is not a big concern, **I will further raise my score to 8**. I will finalize my score during the discussion period with ACs. Thanks to the authors for their patience.
>
> **I would like to also mention, that proposing the alignment on the model parameter posteriors across domains is very novel, to other reviewers and AC.** It offers a new baseline to enhance domain generalization using the core concept from BNN, which is significant to the machine learning community in general.

---

### Official Review · Reviewer_BPVw · 2023-10-31

**Soundness:** 1 poor
**Presentation:** 2 fair
**Contribution:** 2 fair
**Rating:** 3
**Confidence:** 3

**Summary:**

This paper investigates a new approach to domain invariant learning using Bayesian neural networks. In particular, the paper first considers a theorem to show that the invariant posterior of parameters can be implicitly inferred by aggregating posteriors on different training domains. This theorem is under an assumption that the domain invariant information $\mathcal{D}^{c}$ and the domain-specific information $\mathcal{D}^{v}$ are independent. Based on this, the paper proposes a method named Posterior Generalization (PTG) and its lite version to estimate the distribution of the invariant posterior. The proposed method is validated on some benchmarks.

**Strengths:**

- The code is provided and will be public.
- The experimental results are good at least from Table 1.

**Weaknesses:**

- The paper is not very sound as it attempts to consider learning domain invariants using the Bayesian framework on the parameter space. This is different from the existing work where the feature space is considered. However, this may lead to challenging problems as the parameters of neural networks are extremely high-dimensional and unidentifiable. It is extremely difficult to align and aggregate the posteriors in the parameter space. Therefore, the authors propose some ad-hoc solutions in Section 3.3 such as neural networks for different domains should be initialized from the same point, the parameters of the last layer should be shared between neural networks, and the learning rate should be carefully decayed. These solutions are not well-motivated and do not guarantee that the posteriors of parameters are well-aligned and aggregated properly.
- The authors propose PTG-Lite, which is a lite version of the main method. Due to the difficulty of employing Bayesian treatment to the neural network’s parameters, PTG-Lite just uses maximum-a-posteriori MAP solutions. We can see from Table 1 that PTG-Lite performs comparably with PTG. This somehow downgrades the main narrative of the paper which is to propose a Bayesian framework for aggregating posteriors on domains and inferring domain invariant posteriors.
- One of the main contributions is the Theorem 3.1. However, it is trivial. Although the authors claim in the abstract that this theorem relies on relaxed assumptions. However, the assumption of full independence between the domain invariant and domain-specific information is quite strong.
- The authors should experimentally compare with other approaches using Bayesian neural networks such as Xiao et al. (2021) and follow-up works.
- The writing should be improved. There are some citation and grammar typos and sentences that need further clarification. For example, in the second row from the bottom of page 3, what are the parameters?

References

Xiao et al. A Bit More Bayesian: Domain-Invariant Learning with Uncertainty. ICML 2021.

**Questions:**

- In Section 3.1, the paper claims that “we do not need to specify how $\mathcal{D}^{c}$ and $\mathcal{D}^{v}$ are extracted from $\mathcal{D}$”. Is this useful as it would be important to learn which information from the data is invariant or not? Is it possible to interpret the inferred domain-invariant posterior?
- In Equations (5) and (7), what are the expectations with respect to?

---

> ### Author Response · Authors · 2023-11-19
>
> ## Weakness 1
>
> Our major contribution is that we propose a method to estimate $p(\omega|\mathcal{D}^c)$ without $\mathcal{D}^c$. The analysis of parameter distributions has been widely studied so far, and variational inference is a simple way to align parameters. Besides Bayesian theorem, we also solve problems that may occur during implementation. Furthermore, our experiment can prove that our effort works.
>
> ## Weakness 2
>
> Although PTG-Lite is not BNN, its principle is still Bayesian. It is Theorem 3.1, which shows how to get the domain invariant posterior $p(\omega|\mathcal{D}^c)$ with only access to $\mathcal{D}$, that matters, rather than way to approximate $p(\omega|\mathcal{D}^c)$. PTG-Lite uses deterministic values to approximate $p(\omega|\mathcal{D}^c)$, but the key idea is still to estimate the invariant posterior.
>
> ## Weakness 3
>
> We explained why $\mathcal{D}^c$ and $\mathcal{D}^v$ should be independent in Appendix A. The proof of Theorem 3.1 is shown in Appendix B. Although Theorem 3.1 appears to be always true, in fact it holds under the condition that $\mathcal{D}^c$ and $\mathcal{D}^v$ are independent. Besides, Theorem 3.1 shows that we can estimate $p(\omega|\mathcal{D}^c)$ by the posterior given full information, which plays a vital role in the following context.
>
> ## Weakness 4
>
> We didn't report the mentioned approaches because they are not appliable on DomainBed. For fair comparison, we only compare with models on DomainBed.
>
> ## Weakness 5
>
> We will correct typos and improve the writing.
>
> ## Question 1
>
> We don't need to specify which information from the data is invariant. In BNN, we care how to estimate $p(\omega|\mathcal{D})$ rather than what's the data distribution $p(\mathcal{D})$. Similarly, we care how to estimate $p(\omega|\mathcal{D}^c)$, rather than $p(\mathcal{D}^c)$. $\mathcal{D}^c$ means any information that keeps invariant between domains, and $p(\omega|\mathcal{D}^c)$ is parameter posterior distribution that use $\mathcal{D}^c$ for prediction.
>
> ## Question 2
>
> These are typos. $E[q(\omega|\theta_i)]$ means $E_{q(\omega|\theta_i)}[\omega]$, and $VAR[q(\omega|\theta_i)]$ means $VAR_{q(\omega|\theta_i)}[\omega]$.

---

> > ### Comment · Reviewer_BPVw · 2023-11-22
> > **Official Comment by Reviewer BPVw**
> >
> > Thanks the Authors for the rebuttal. However, the rebuttal did not address satisfactorily all my concerns. Therefore, I would keep the current score.

---

### Meta-Review · Area_Chair_AbzB · 2023-12-08

**Metareview:**

As pointed out by multiple reviewers, one of the main concerns that has been not properly discussed or addressed is the motivation of the chosen invariance criterion in this work -- compared with prior works that focus on learning invariant features, this manuscript focuses on learning invariant posterior distributions over different groups. However, it is completely unclear why is this a good criterion of invariance to consider. Is there any formal connection between this criterion with invariant features? How does this criterion lead to domain generalization?

Apart from the key motivational issue, several other issues need to be addressed before this manuscript can be considered for publication. For example, as also pointed out by multiple reviewers, the presentation of the work could be significantly improved, in terms of both references, grammar typos, and clear definitions of technical terms used throughout the manuscript. I would also suggest the authors tone down the claims made in the manuscript (consider changing Theorem 3.1 to a proposition instead since it's indeed quite straightforward to prove). There are quite a few other major technical concerns as well, and the authors were not able to provide convincing responses to these concerns during the rebuttal period.

The reviewers have provided many constructive feedback and suggestions for improvement. I would suggest the authors carefully address all the comments and suggestions made by the reviewers in the next iteration of the manuscript.

**Justification For Why Not Higher Score:**

All the reviewers share serious concerns about the paper and the authors were not able to address them during the rebuttal. The manuscript also needs significant polish before publication.

**Justification For Why Not Lower Score:**

N/A

---

### Decision · Program_Chairs · 2024-01-16

Reject